

# Evaluation of the interactive stratospheric ozone (O3v2 module) for the E3SM version 2 Earth System Model

Qi Tang[1], Michael J. Prather[2], Juno Hsu[2], Daniel J. Ruiz[2], Philip J. Cameron-Smith[1], Shaocheng Xie[1], Jean-Christophe Golaz[1]

[1]Lawrence Livermore National Laboratory, Livermore, CA 94550, USA
[2]Departments of Earth System Science, University of California, Irvine, Irvine, CA 92697, USA

*Correspondence to*: Qi Tang (tang30@llnl.gov)

**Abstract.** Stratospheric ozone affects climate directly as the predominant heat source in the stratosphere and indirectly through chemical feedbacks controlling other greenhouse gases.  The U.S. Department of

Energy's Energy Exascale Earth System Model version 1 (E3SMv1) implemented a new ozone chemistry module that improves the simulation of the sharp tropopause gradients, replacing a version based partly on long-term average climatologies that poorly represented heating rates in the lowermost stratosphere.  The new O3v2 module extends seamlessly into the troposphere and preserves the naturally sharp cross-tropopause gradient, with 20-40% less ozone in this region.  Additionally, O3v2

enables the diagnosis of stratosphere-troposphere exchange flux of ozone, a key budget term lacking in E3SMv1.  Here, we evaluate key features in ozone abundance and other closely related quantities in atmosphere-only E3SMv1 simulations driven by observed sea surface temperatures (SSTs, years 1990-2014), comparing with satellite observations and the University of California, Irvine chemistry transport model (UCI CTM) using the same stratospheric chemistry scheme but driven by European Centre

forecast fields for the same period.  In terms of stratospheric column ozone, O3v2 shows improved mean bias and northern mid-latitude variability, but not quite as good as the UCI CTM.  As expected, SST forcing does not match the observed quasi-biennial oscillation, which is mostly matched with the UCI CTM.  This new O3v2 E3SM model retains mostly the same climate state and climate sensitivity as the previous version, and we recommend its use for other climate models that still use ozone

climatologies.



# 1 Introduction

Accurate simulation of past climate evolution and projections of future climate rely on correct representation of the greenhouse gases. This can be a challenge for atmospheric ozone, which has large critical gradients and requires chemistry-transport modelling. The importance of two-way interaction
between chemically active climate compounds and climate change has been recognized in previous studies as occurring through changes in radiation, temperature, dynamics, and the hydrological cycle (e.g., Isaksen et al., 2009; Raes et al., 2010; Dietmüller et al., 2014; Nowack et al., 2015). These feedbacks can either dampen or exacerbate $CO_2$-driven warming. Climate change studies through Coupled Model Intercomparison Projects (CMIPs) (e.g., Taylor et al., 2012; Eyring et al., 2016) have
generally chosen to prescribe greenhouse gas abundances based on historical observations or projected emissions with simple biogeochemistry box models. This approach works for the well-mixed greenhouse gases but is a poor approximation for ozone. Ozone is a short-lived reactive gas, is not directly emitted, has many sources and sinks in the atmosphere, and maintains sharp gradients at dynamical boundaries. Running a full atmospheric chemistry model for ozone within a climate model
is costly, often prohibitively, and thus most climate simulations adopt mean climatological distribution based on present-day observations or some external chemistry-climate model simulation. The problem with this approach is that the externally prescribed ozone never aligns with the model's dynamical boundaries (i.e., tropopause, Antarctic stratospheric vortex) and thus heating by ozone is deposited across these boundaries, tending to weaken them, altering the climate simulation. Thus, many Earth
system models (ESMs) are now incorporating some form of interactive ozone chemistry.

The U.S. Department of Energy's (DOE) Energy Exascale Earth System Model version 1 (E3SMv1) (Golaz et al., 2019; Rasch et al., 2019) implemented chemistry-climate interactions through stratospheric ozone by incorporating linearized chemistry (Linoz v2, Hsu and Prather, 2009), which
could be included with little impact on the computational cost of climate simulations. Linoz v2 calculates the first-order Taylor expansion terms for the stratospheric ozone production and loss based on local temperature, local ozone abundance, and the overhead ozone column, and is tabulated for different levels of greenhouse ozone-depleting gases. Linoz has been applied in various chemistry



transport models (CTM), including the University of California, Irvine (UCI) CTM, and produces a reasonable ozone climatology, including seasonal and interannual variability (McLinden et al., 2000; Hsu et al., 2005; Hsu and Prather, 2009). In the first use of Linoz in E3SMv1, the O3v1 module prescribed tropospheric ozone based on decadal monthly zonal mean latitude-by-pressure data from

5 another model and calculated stratospheric ozone interactively with Linoz v2. O3v1 resulted in unphysical ozone distributions about the tropopause, i.e., when the tropopause rose relative to the climatological tropopause, the ozone climatology overwrite would place large stratospheric abundances into tropospheric air masses and these errors were not symmetrical. Similar problems occurred in the vicinity of sub-tropical and polar jets. Altogether, these errors have an uncertain climate impact and

10 thus, here, we implement an improved O3v2 ozone module in E3SMv1 and perform a more comprehensive evaluation of the ozone simulation, comparing with satellite observations and with the UCI CTM running the same O3v2 chemistry. O3v2 also enables ready diagnostics of stratosphere-troposphere exchange flux of ozone. Furthermore, we examine how the O3v2-O3v1 changes in both mean climate and climate sensitivity of E3SMv1.

Section 2 describes the model, the simulations, and the observations. The E3SM model performance of stratospheric ozone against satellite observations and including UCI CTM simulations is shown in Section 3. A detailed look at O3v2 versus O3v1 differences, including present-day climate simulation and climate sensitivity is given in Section 4. Discussion and conclusions are in Section 5.

## 20 2 Experimental design

### 2.1 Model description

The overall description of E3SMv1 is provided in Golaz et al., (2019). The atmospheric component (EAM version 1) of E3SMv1 is described in Rasch et al., (2019) and Xie et al., (2018). All the E3SM simulations in the present study are performed with EAMv1 forced by monthly mean SSTs at the

25 standard 1° horizontal resolution and 72 vertical layers, extending from the surface to 60 km (~ 0.1 hPa) with a 600 m vertical resolution near the tropopause (see Fig. 1 of Xie et al., 2018). The first EAMv1





ozone package (termed O3v1) uses a prescribed decadal monthly mean climatology from the input4MIPS ozone data set v1.0 (Hegglin et al., 2016) in the troposphere, but uses the prognostic linearized ozone chemistry scheme (Linoz v2) (Hsu and Prather, 2009) in the stratosphere. Linoz calculates the stratospheric ozone net tendency with its first-order Taylor series expansion as a function

of local ozone mixing ratio, local temperature, and overhead ozone column. The linearized production and loss coefficients are updated for E3SMv1 using the greenhouse gas (GHG) concentrations from the input4MIPS GHG historical data set v1.2.0. Following Cariolle et al. (1990), Linoz uses a parameterization for chlorine-induced ozone depletion based on temperature and sunlight thresholds intended to mimic chlorine activation on polar stratospheric clouds (PSCs) at cold temperatures and the

ensuing rapid photochemical loss of ozone.  This model has proven robust and reasonably accurate. The combined troposphere-plus-stratosphere ozone profile is generated by the combined Linoz chemical tendencies and EAM tracer transport throughout the atmosphere but is then overwritten below the instantaneous EAM tropopause with the input4MIPS climatology even when that climatology has stratospheric values.  The ozone profiles are passed to the radiative transfer module for radiative heating

calculations.

The O3v1 package has some clear weaknesses. Overwriting the EAM tropospheric values every model time step with the monthly climatologies misses the ozone variability associated with the regular ridge-trough tropopause changes, obscuring the sharp cross-tropopause gradient in ozone and ozone heating

rates.  More importantly, O3v1 assigns stratospheric high-$O_3$ concentrations to tropospheric air when the EAM tropopause rises above the monthly climatology in the prescribed data set.  This systematic overestimation of ozone near the tropopause has unknown climate impact. In the present study, we correct these problems with the O3v2 chemistry module by replacing the tropospheric overwriting with a tropospheric tracer that is passive except in the lowest four layers (below 1 km altitude) it is removed

with a 48-hour e-folding decay to 30 ppb (parts per billion by mole fraction).  The choice of 30 ppb gives Linoz a tropospheric ozone mass similar to full chemistry models and observations (Ziemke et al., 2019). Therefore, O3v2 is able to interact with tropopause changes and maintain the naturally sharp ozone gradient across the tropopause.  Linoz v2 was developed for the UCI CTM and shows



consistently reliable stratospheric ozone simulations (Hsu and Prather, 2009). It has been implemented in other models such as European Centre-based CTMs (Aschmann et al., 2009), the CESM-CAM-Superfast climate model used in ACCMIP (Lamarque et al., 2013), and current versions of GEOS-Chem (Murray et al., 2012; Hu et al., 2018).

Additionally, the lower boundary sink introduced by O3v2 provides a self-consistent diagnostic for the stratosphere troposphere exchange (STE) flux of ozone, a major tropospheric ozone budget term, which cannot be diagnosed in O3v1. Many of the new chemistry-climate studies are now including this methodology, i.e., the use of a stratosphere-only ozone tracer, called $StratO_3$, to calculate the STE ozone

flux (Liu et al., 2020).

The O3v1 module was originally set to match the observed Antarctic ozone hole using a PSC temperature threshold of 193 K in EAMv1. This value is less than the 195 K threshold in Cariolle et al. (1990) and the 199K threshold used in the UCI CTM because EAMv1 with O3v1 had a much colder

winter pole than the other models. When EAMv1 is paired with O3v2, the Antarctic winter pole is warmer, and we find that a PSC threshold of 197.5 K represents the best ozone hole performance (see Section 3.3).

As a global climate model forced with observed SSTs, EAM inevitably has difficulties in matching the

meteorological conditions of the period, especially in the stratosphere ranging from the jet positions, to interannual winter warmings, to the quasi-biennial oscillation (QBO). Fortunately, we can use the UCI CTM to provide a reference check because it runs the same O3v2 chemistry package and uses European Centre 3-hourly forecast fields from their T159L60 Integrated Forecast System (1.1° horizontal resolution) over the same time period (Prather et al., 2017). The UCI CTM does not necessarily have

the correct transport since all re-analysis or forecast wind fields have their own uncertainties, especially when it comes to residual transport that controls the ozone distribution. It would be interesting to run EAM as an offline CTM driven by re-analysis winds, but the existing EAM nudging capability (Sun et



al., 2019; Tang et al., 2019) does not support this application. More importantly, our goal here is to test the free-running climate model.

## 2.2 Model simulations

The model simulations analysed in this study are summarized in Table 1. The control simulation uses

one of the three Atmospheric Model Intercomparison Project (AMIP) simulations (see Golaz et al., 2019 for more details) forced with prescribed sea surface temperatures (SST) and sea ice concentrations following the CMIP6 DECK protocol (Eyring et al., 2016). The control simulation is performed for years 1870-2014. We configure the EAM O3v2 test run with the same AMIP settings as the control but with O3v2 modifications described above. We initialize the O3v2 simulation with the beginning of year

1990 conditions of the control and run it through the end of 2014. For the analysis here, we focus on the last 20 years of the EAM O3v2 run and skip the first 5 years as spin-up. The UCI CTM hindcast simulation covers years 1990-2017. Since the UCI CTM is driven by forecast winds initialized with observationally assimilated data, it is capable of simulating time-specific observations, and we compare with ozone observations for those simulated years. One additional pair of 5-year O3v2 AMIP

simulations are carried out to diagnose the climate sensitivity (following Cess et al., 1989). One of the pair prescribes the SST and sea ice concentration to represent current climate conditions. The other simulation is identical except for increasing the SST uniformly by 4 K. More details about the E3SMv1 Cess configuration are documented in Caldwell et al., (2019).

## 2.3 Evaluating models versus observations

The observational metrics used here are (i) monthly zonal mean stratospheric column ozone (SCO); (ii) similarly monthly averaged ozone profiles in the stratosphere; and (iii) daily geographically resolved total column ozone (TCO) following the evolution of the Antarctic ozone hole. To avoid confounding potential errors in the stratosphere with those in the troposphere, we take the uncommon approach of

comparing only SCO. The SCO data (i) are derived from Ziemke et al.'s (2006, 2019) work that merges total column ozone data from the Ozone Monitoring Instrument (OMI) with stratospheric profile data

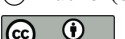



from the Microwave Limb Sounder (MLS) to calculate a tropospheric column ozone. Both instruments are on the NASA Aura satellite (Schoeberl et al., 2004). The Ziemke data set is geographically resolved, but here we use only the zonal monthly mean. The zonal mean ozone profile data (ii) are provided in the MLS level 3 gridded data set ML3MBO3 V004 (Schwartz et al., 2020). We linearly

interpolate the model results to the coarser observational grids when calculating model-observation differences. The daily TCO data (iii) are collected by the Total Ozone Mapping Spectrometer (TOMS) on the NASA/NOAA Nimbus-7 satellite (McPeters et al., 1996), the OMI instrument on the Aura satellite, and the Ozone Mapper and Profiler Suite (OMPS) on the NASA's Suomi National Polar-orbiting Partnership (NPP) satellite (Flynn et al., 2018). The missing daily TCO data due to bad orbits

and polar night are filled with the assimilated data from the Modern-Era Retrospective analysis for Research and Applications (MERRA) (Rienecker et al., 2011). Based on these daily TCO data, the NASA Ozone Watch website (https://ozonewatch.gsfc.nasa.gov, accessed on 29 May 2020) compiles the daily records of the Antarctic ozone hole area (defined as TCO < 220 DU) and minimum TCO. In this study, we use the data obtained from the Ozone Watch website to evaluate model simulations.

Table 2 lists the details of the observational data used here.

## 3 Performance metrics for stratospheric ozone simulations

One of our goals is to establish a set of standard climate model metrics that address the simulation of

stratospheric ozone. Thus, it is important to separate the ozone column data (in DU, Dobson Units, milli-cm-Amagats) into stratosphere and troposphere (see Ziemke et al. (2019) for derivation and analysis of the tropospheric column). This is not typically done, but it is important since tropospheric ozone has its own driving forces for both trends and interannual variability. In fact the trends in ozone column over the past decades appear to be driven by tropospheric ozone (Gaudel et al., 2018). We also

develop metrics based on the profiles of ozone and the evolution of the Antarctic ozone hole (for which we use total column ozone). The last ozone metric that we would like to use is the stratosphere-





troposphere exchange flux since it is an important link between the two ozone reservoirs. Unfortunately, we have no direct observations and rely on model-model comparisons.

## 3.1 Stratospheric column ozone

The SCO observations are limited to the range 60°S to 60°N where the best satellite observations relying on sunlight are year-round, and our performance metrics follow this limit. The multi-year average annual cycle of SCO (zonal means, month by latitude) are shown in Figure 1 for the observations (OMI+MLS) and models (UCI CTM and both EAM versions O3v1 and O3v2). The

multi-year averages include the specific years 2005-2017 for OMI+MLS and UCI CTM and SST-forced years 1995-2014 for both EAM versions. The model simulations are reasonable but with obvious errors: UCI CTM is systematically low everywhere but matches the pattern; EAM versions are excellent in the tropics but too great at high latitudes. The lower overall SCO outside the tropics seen in O3v2 versus O3v1 is a closer match to the observations. In Figure 1 we also show the difference plot of EAM

O3v2 minus O3v1 (extended to 90°S - 90°N). Outside the tropics, the O3v2 SCO is consistently 15-30 DU less than that of O3v1, a direct result of the O3v1 error in overwriting $O_3$ in the lower stratosphere and upper troposphere.

Climate models often capture the mean better than the variability. Thus, we create a metric based on

the interannual anomalies in the annual cycle as a function of latitude (STD of the SCO in DU, Figure 2). This allows us to focus on interannual variability in the tropics, presumably QBO-related, and in mid-to-high latitudes, presumably wintertime polar variations. The observed STD/SCO are for 2005-2017 (black solid line), and for the models we present two different 13-year periods to address the uncertainty in calculating STD from such a short record. EAM versions use years 1995-2007 (dashed)

and 2002-2014 (solid); while UCI CTM uses specific years 1992-2004 (dashed line) and 2005-2017 (solid). All model results reproduce the general pattern in observations: peak QBO-like variability (~7 DU) in the core tropics, a minimum (3 DU) at 15° latitude, then steadily increasing back to tropical levels by 60°. Thus STD/SCO provides a second test of the overall stratospheric circulation. All





simulations overestimate the STD of anomalies near the Equator, while fluctuating around the observation over extra-tropics, especially in the Northern Hemisphere (NH). UCI CTM STD gravely overestimates in the tropics and is consistently higher at all latitudes. Both EAM versions match well in the tropics and northern latitudes, but underestimate variability in the southern latitudes. There is no

clear separation of O3v2 and O3v1 with this metric. Except for the 10-15% higher STD in the tropics and southern latitudes for the 1992-2004 UCI CTM period, the different model periods show modest difference and thus the observational period is probably adequate for this metric. The jump in the long-term UCI CTM STD may be due to changes in the wind-driven circulation caused by the switch in satellite data used in the assimilation from the Solar Backscatter Ultraviolet Radiometer to MLS and

OMI (Wargan et al., 2017).

We present a Taylor diagram for the mean annual SCO cycle [1] and the STD/SCO [2] in Figure 3a. What is being evaluated in [1] is the model simulation of the 2D area-weighted pattern in Figure 1; and in [2], the 1D (area weighted) line plots in Figure 2. The observed pattern is plotted at the (1,0)

reference point. For [1], all models simulate high correlations (> 0.95), suggesting well-captured annual cycle. We expect all stratospheric chemistry models will do very well on this test because, while this specific metric has not been used before, all modelers have been using the 'eyeball' metric for decades when comparing with total column ozone in figures similar to Figure 1. The UCI CTM scores slightly better because of the high-latitude SCO. For [2], the root-mean-square (RMS) errors (represented by

the radius of the arcs centred on the (1,0) point) are similar for UCI CTM and O3v1 (radius of 0.75 standard deviations), and best for O3v2 (radius of 0.50). There is no clear explanation of this improvement in O3v2 except perhaps that the reduction in lower stratospheric $O_3$ changed the wave propagation and variability seen in the southern latitudes. These two metrics are clearly independent and highlight different aspects of the chemistry-climate system, where even a model running the same

O3v2 chemistry with assimilated-forecast meteorology does not always perform better.





## 3.2 Stratospheric ozone profiles

The distribution of $O_3$ within the stratosphere is important not only as diagnostic of chemistry and transport but also the driver of stratospheric heating. We thus choose a metric based on the Aura MLS observations of the monthly zonal mean cross section (latitude by pressure) of ozone abundance (ppm, mole fraction in parts per million) as shown in Figure 4. Our ozone profile metric includes all 12 months plus the annual mean, but only June and October plus the annual mean in Figure 4. We avoid the lowermost stratosphere where zonal variability is large and restrict ourselves to a pressure range of 100 to 0.2 hPa (approximately 16 to 50 km altitude). This metric has been a standard test for 2D and 3D stratospheric chemistry models for decades. The model goal (not usually quantified) was to get the peak 10 ppm at 10 hPa in the tropics and the slightly upturned contours (i.e., at 5 hPa the 6 ppm contours extend over a wider latitude range than at 20 hPa). Overall the models match the observed patterns, including the odd seasonal upward shift of contours in the winter (60°S in June and 60°N in October). This test emphasizes the region where photochemistry is active (sunlit latitudes) and ozone is in a quasi-steady state and little influenced by transport. Since both UCI and EAM are using the same chemistry module they should give nearly identical results in this test. The chemistry depends somewhat on temperature and that can explain the slightly larger peak tropical $O_3$ in EAM versions. Poleward of 60°, transport plays a more important role and we see the differences between UCI and EAM.

In terms of Taylor diagrams, this metric collapses to a small region indicating excellent performance (Figure 3b). Correlations are close to 1.0 for all simulations, indicating excellent pattern agreement. Variances are underestimated by the models, implying that the linearized chemistry is based on a more uniform set of background conditions than those occurring in the stratosphere, and this is to be expected. The 20% $O_3$ differences between O3v2 and O3v1 in lowermost stratosphere have little impact on this metric as expected. If all models used Linoz chemistry, this would be a not-useful metric, but since many have their own independent chemistry modules, we expect this to be a useful check.



The interannual variability of these monthly mean profiles is more difficult to reproduce. Here we are not trying to match specific year-to-year changes but calculating a monthly latitude-by-pressure map of the 13-year record of standard deviations of $O_3$ abundance in ppm at each point. The Taylor diagram

for these data (Figure 3c) shows that all models become worse than they did for the climatological mean, specifically with smaller correlations and smaller RMSE. The obvious explanation is the interannual variations in the middle stratospheric consist of both temperature (mapped reasonably into $O_3$ variations by Linoz) and chemical variations (not included in Linoz). This metric is driven by the large interannual MLS variations in the tropical middle stratosphere (not shown). UCI CTM has the

closer match to MLS observations than either EAM versions, both in terms of STD and correlation. This metric is a tough one and clearly separates the two models, but we will need to add some other models to see how well it works outside of Linoz chemistry.

### 3.3 Antarctic ozone hole

The statistics of the evolution of the Antarctic ozone hole since 1990 have been driven primarily by dynamical variations because the chlorine levels driving ozone depletion inside the Antarctic winter vortex have evolved slowly. We thus use the daily ozone-hole diagnostics from the NASA Ozone Watch website for our metric (see Figure 5). The two quantities are (i) the area (in millions of km$^2$) with less than 220 DU in total column ozone, and (ii) the minimum total column ozone (in DU). The

thick lines in Figure 5 represent the multi-year average of the daily values, and the shaded areas indicate the range of ±1 standard deviation about this average. We show results for the observations and O3v2 and O3v1. The 20-year time series of ozone column for O3v1 (Fig 5a) clearly shows the regular occurrence of the ozone hole with suitable variability like the minimal ozone hole in 2003. The parallel difference plot of O3v2 minus O3v1 (Fig 5d) shows the O3v1 errors in the lowermost stratosphere as

large wintertime biases of excess ozone column. In O3v2 there is more interannual variability in the ozone hole with more frequent minimal values like 1995 and 2001. The UCI CTM did not record daily ozone values and cannot be assessed here.





The O3v1 ozone-hole area is generally about 30% smaller than the observation (Fig. 5b) and also about 30 DU less deep than observed from August through November (Fig 5c). O3v2 clearly matches the observations better, both in terms of area and minimum value. The clear improvement with O3v2 is the onset of the hole where O3v1 shows almost a two-week delay but O3v2 matches the observations. We must be careful in judging O3v1, because its ozone-hole performance could possibly be tuned with a better PSC temperature threshold.

Overall, the onset and duration of the ozone hole seem relatively unchanged with different E3SM configurations. The Taylor metrics are similar for both (Fig. 5d). We infer that the large-scale dynamical conditions are similar in both EAM versions and remain relatively isolated from the ozone-hole chemistry, unlike the changes in dynamics caused by the O3v1-O3v2 changes near the tropopause discussed below. The Antarctic stratosphere comes out of wintertime with PSC activation of chlorine between 14 and 22 km altitude, thus the altitude range of PSCs will likely control the evolution of the ozone hole. This would require a separate diagnostic.

## 3.4 STE ozone flux

With O3v2 the E3SM model is able to diagnose the STE ozone flux ($TgO_3$ per yr), which is a key budget term for tropospheric $O_3$. We can place constraints on the global mean ozone flux based on proxy relationships with other trace gases, and this approach gives us a broad range of 400-600 $TgO_3$/yr (Murphy and Fahey, 1994; McLinden et al., 2000; Olsen et al., 2001, 2004; Hsu et al., 2005). Unfortunately, using satellite data to resolve the STE flux is difficult. For example, Hsu et al. (2005) identified a large apparently isolated column ozone anomaly (1.7 Tg-$O_3$) as seen by satellite during an STE event in the eastern Pacific; the UCI CTM was able to match the anomaly, but in following that stratosphere-troposphere folding event for several days, they found that most of the $O_3$ mass remained stratospheric and only about 20% was mixed into the troposphere. Tang and Prather (2012) evaluated the possibility of quantifying the STE ozone flux with independent ozone measurements from the four





Aura instruments. They concluded that it would be challenging to integrate the flux only based on the satellite observations. Thus, for STE flux as a function of latitude and month, we compare across models, and in this case with the UCI CTM. If we collect enough different models with enough similar results, then maybe we can build a Taylor diagram using an ensemble mean reference case.

In O3v2 the STE ozone flux is calculated as the tropospheric loss, which is set to the lowest four layers. It is averaged over latitude and month. In the UCI CTM, the STE flux is diagnosed at the tropopause defined by an e90 tracer (Prather et al., 2011) and is able to resolve the STE fluxes across multiple tropopauses in the same column (Hsu et al., 2005; Tang et al., 2013; Hsu and Prather, 2014).

10 Comparing these two STE fluxes as monthly zonal means is appropriate as there is no apparent bias in the two methods and the time lag for transport from tropopause to surface is less than a month (Jacob, 1999).

Figure 6 illustrates the multi-year seasonal cycle of the STE ozone flux in each hemisphere for O3v2

15 (red) and UCI (green). The annual mean values are similar in both models: in the NH (solid lines), 215 $TgO_3$/yr for UCI and 215 for O3v2; and in the SH (dashed), 190 and 170. The seasonal amplitude for UCI is, however, twice as large as that for O3v2. For O3v2 the NH STE flux (solid lines) peaks in May and bottoms in Dec, while for UCI, the peak extends to Jun, and the minimum occurs much earlier in Sep-Oct. The SH flux has generally the opposite phase to the NH flux, but here the two models separate

20 in phase by about 4 months. None of these phase differences can be accounted for by differences in the methods. The shaded area about each line represents the ±1 standard deviation about the multi-year daily average, and both models have similar year-to-year variability. Values here, ~400 $TgO_3$/yr, fall at the lower end of the constrained global mean flux. E3SM O3v2 will now be able to contribute STE ozone fluxes to future MIPs (Young et al., 2018).

## 4. Climate changes from O3v1 to O3v2





The seemingly small changes from O3v1 to O3v2 had a surprisingly large impact on the lower
stratosphere (Fig. 7), with O3v2 having about 20% less ozone in the lower stratosphere, but hardly any
change in the troposphere and small changes in the mid-stratosphere (not shown). In this section, we
will examine the changes between O3v1 and O3v2 in greater details.

## 4.1 Changes in UT/LS

Figure 7 shows the pressure-by-latitude O3v1 ozone, as well as, O3v2-O3v1 differences in the annual
mean, and June and October in the upper troposphere/lower stratosphere (UT/LS) region (50-400 hPa).
The tropopause pressure (green lines) from the O3v1 simulation is overlaid on the contours to facilitate
comparisons. Panels a-c illustrate the typical UT/LS ozone pattern: ozone decreases from the middle
stratosphere to the lower stratosphere and with a sharp gradient across the tropopause. Panel c shows the
ozone hole depletion at 70 hPa over the South pole in October. Compared to O3v1, O3v2 simulates less
ozone throughout the UT/LS region except at the lower stratosphere over SH high latitudes. The
reduction of O3v2 ozone is consistent with the lack of high-frequency tropopause variability in the
O3v1 prescribed ozone climatology data as described in Section 2.1. The positive O3v2-O3v1 change at
the SH high latitudes is caused by more wave activity and meridional transportation from the middle
latitudes to the polar region. The mechanism of this ozone increase is further investigated with the
composite data from years when the O3v2 ozone holes are substantially weaker with the same PSC
temperature threshold as O3v1 (Fig. A1). Strong ozone enhancement occurs at the SH high latitudes
(Fig. A1a) along with temperature increases (Fig. A1b) and polar vortex weakening (Fig. A1c). It
appears that the heating changes near the tropopause lead to changes in its stability as shown by the
buoyancy frequency squared ($N^2$) (Fig. A1e), altering the wave propagations as a valve: the enhanced
vertical gradient of $N^2$ suppresses wave propagation from the tropopause to the stratosphere over SH
high latitudes (Chen and Robinson, 1992; Simpson et al., 2009), whereas the decreased $N^2$ gradient at
SH middle latitudes tropopause facilitate the wave propagation carrying the poleward heat flux. The
mean Eliassen-Palm (E-P) flux and its differences in divergence (Fig. A1f) present a consistent picture
as the $N^2$. Similar thermal-dynamical responses to the heating changes near the tropopause are reported





by Hsu et al., (2013) when changing the ozone production from $O_2$ photolysis in the lower tropical stratosphere.

The radiative transfer code in E3SM takes into account the ozone changes (Rasch et al., 2019) and thus responds with different heating profiles. The total net heating (shortwave + longwave) results from the E3SM simulations are shown in Fig. 8 for the UT/LS region. The O3v2 causes slight (up to a few percent) net cooling around the tropopause at all latitudes (except at SH high latitudes in austral spring-summer time) and net warming in the lowermost stratosphere in the annual means. When separating the heating profile changes into shortwave and longwave (not shown), the cooling signal near the tropopause is a combination of the cooling in both shortwave and longwave, whereas the warming in the lowermost stratosphere is because the warming in the longwave dominates the cooling in the shortwave. The warming near the tropopause at SH high latitudes is mainly caused by the shortwave absorption.

The cooling near the tropopause and warming above generally lead to a higher tropopause defined by the temperature lapse rate (Fig. 9). The tropopause changes are greater in the extra tropics when compared to the tropics, and even larger poleward of 60 degrees where tropopause variability is greater. The O3v2 tropopause is generally higher by up to 10 hPa than that of O3v1 except during a few months over the poles (i.e., July and August in the Antarctic and December in the Arctic).

## 4.2 Climate impact

### 4.2.1 Mean climate

Model development often experiences the dilemma of improving some parts of the model performance at the cost of deterioration of other parts. The more physical representations of processes do not necessarily lead to better model performance against observations. Therefore, it is critical to ensure that the O3v2 scheme does not cause significant degradation of the simulated mean climate as well as the climate sensitivity. Here we apply the same diagnostic to examine the overall climate performance as



used in the E3SMv1 overview papers (Golaz et al., 2019; Caldwell et al., 2019). In this diagnostic (Figs. 10 and 11), we compute the uncentered RMSE relative to observations for the E3SM models and 30 CMIP5 AMIP models with the PCMDI Metrics Package (PMP) (Gleckler et al., 2016). The E3SM simulations cover the period of 1995-2014, while the CMIP5 ensemble years 1981-2005. The spatial

RMSE of the annual and 4 seasonal averages for 9 variables are presented as boxes and whiskers for the CMIP5 ensemble and blue dots for O3v1 and red dots for O3v2. Smaller numbers mean better simulations.

For the global comparisons (Fig. 10), at the top-of-atmosphere (TOA) the radiation variables (panels a-

c) are similar or slightly better for O3v2 than for O3v1. At the surface, the precipitation (panel d), surface air temperature over land (panel e), and zonal wind stress over ocean (panel f) remain relatively unchanged, except slight degradations for surface air temperature over land during March-May and for ocean zonal wind stress for December-May. At different levels, the 200-hPa and 850-hPa zonal wind and 500-hPa geopotential height show small changes to both directions.

Since the O3v2 configuration changes the PSC ozone loss T threshold, it is expected to have larger impact over the SH high latitudes. We further analyse the climate impact at 50S-90S (Fig. 11). While other TOA radiation fields are alike, the longwave cloud radiative effect becomes worse during December-May, suggesting changes in the high clouds or the phase partitioning of mixed phase clouds

during this period. The surface precipitation is similar for all seasons with the exception of June-August when the O3v2 result is slightly improved. Greater changes are found in the thermo-dynamical fields due to the climate-dynamics interactions related to the polar vortex and the ozone hole discussed in the previous section. The changes to these fields are mostly towards deteriorating the simulation.

### 4.2.2 Climate sensitivity

Besides simulating the mean climate state, quantifying the climate sensitivities to various forcings is another fundamental goal of climate models, providing valuable insights to climate change especially to future climate projections. Nowack et al., (2015) shows a strong negative climate feedback when



including the interactive stratospheric ozone chemistry in an Earth system model. The negative feedback reduces the climate sensitivity and is mainly caused by the changes in longwave radiation associated to the Brewer-Dobson circulation-driven reduction of ozone and water vapor at the tropical lower stratosphere, and cirrus cloud changes. Since the tropospheric ozone is prescribed in E3SMv1

with O3v1, undermining the degrees of freedom of interactive stratospheric chemistry, the tropopause changes to the quadrupling of CO2 lacks the proper ozone responses near the tropopause, leading to uncertainties in the climate sensitivity derived from such a simulation (Golaz et al., 2019). We are able to quantify this impact by comparing the climate sensitivity of using O3v1 versus O3v2.

We opt to perform the Cess experiment (Cess et al., 1989) to compute the net climate feedback parameter ($\lambda$) to facilitate the comparisons with the published sensitivities of various E3SM configurations (Caldwell et al., 2019). The net climate feedback parameter is defined as the change in the TOA radiative imbalance caused by 1 K change in the global mean surface air temperature. The Cess experiment consists of a 5-year AMIP control simulation and a 5-year AMIP test simulation that is

identical to the control but with the SST increased uniformly by 4 K. Table 3 lists the $\lambda$ numbers calculated from the Cess experiments from the present study and from Caldwell et al. (2019). The high-resolution (0.25 degree) and low-resolution with high-resolution parameters configurations are denoted as HR and LRtunedHR, respectively. With more physical representation of ozone interactions near the tropopause, the O3v2 leads to a slightly greater (in the magnitude) $\lambda$ than the O3v1. The O3v1-O3v2

sensitivity change is much smaller than changes driven by altering the horizonal resolution or physical parameters. This result suggests that the high E3SMv1 climate sensitivity (defined proportional to the reverse of $\lambda$) is not related to the O3v1 deficiencies. Similar to Nowack et al., (2015), our O3v2 simulations also show ozone decrease around the tropopause and tropopause lifting at the tropics (see Section 4.1). These consistent changes likely hint that the same mechanism (Brewer-Dobson

circulation) is responsible for the E3SM sensitivity change.

In summary, the E3SM climate representation is slightly altered (some slight improvements, but more small degradations) with the new O3v2 scheme compared to the default O3v1 scheme. Nevertheless,





the changes do not affect the fidelity of either the mean climate or the climate sensitivity of the simulation.

## 5. Discussion and conclusions

The E3SMv1 model has built capabilities for climate modelling of the water cycle, biogeochemistry,

and cryosphere (Golaz et al., 2019; Rasch et al., 2019). In a next-stage development that focuses on atmospheric chemistry, we re-examined the current model's treatment of ozone (O3v1) and found some errors in the design that led to unphysically large ozone abundances in the lowermost stratosphere. We corrected this with a new ozone module, O3v2, and document the results here. We also built some performance metrics for stratospheric ozone that will become a standard part of E3SM diagnostics.

Running these metrics with O3v1, O3v2, and the UCI CTM was informative. The UCI model, which uses the same $O_3$ chemistry as O3v2, but is driven by ECMWF forecast fields, produced only slightly better results, indicating the stratospheric transport in E3SMv1 is reasonably represented. This is somewhat surprising given that the stratospheric transport was not closely evaluated and tuned for. By adjusting the extremely delicate temperature threshold for PSC formation and thence activation of rapid

chlorine-driven ozone depletion, the Antarctic ozone holes produced in all three models are close to that observed. The STE ozone flux resolved by month-latitude in O3v2 is notably different from that in the UCI model, but we have no observations to evaluate the two, except for the global mean flux where the two models agree.

As we have learned, with changes to physics modules in an ESM, there are climate surprises. The reduced heating in the mid- and high-latitude lower stratosphere changed the stability ($N^2$) of the region and altered the transmission of waves into the stratosphere, which in turn altered the residual circulation and stability of the Antarctic springtime stratospheric vortex. This new circulation led to early breakup of stratospheric vortex and weaker ozone holes in several years. Such phenomena are similar to those

seen in experiments where changing ozone production in the lower tropical stratosphere caused a dramatic shift in high-latitude winter variability (Hsu et al., 2013). The temperatures in the Antarctic





winter stratosphere shifted warmer in O3v2, and thus to maintain the same region of ozone depletion as O3v1, we had to adjust the PSC threshold temperature from 193 K to 197.5 K.

Beyond just ozone, we reviewed other climate diagnostics from EAMv1 O3v2, applying the same
diagnostics as in the E3SMv1 overview papers (Golaz et al., (2019), Rasch et al., (2019), and Caldwell et al., (2019)). Globally, the O3v2 and O3v1 climate are almost identical. Over the SH high-latitudes, however, the changes are greater for some climate variables connected with the ozone hole changes. In terms of climate sensitivity, the good news is that the O3v2 version is within 2% of the original O3v1; whereas alternate E3SMv1 configurations with increased resolution or physical tunings show distinctly
different (6-12%) climate sensitivities. So we must accept good fortune: we identified and fixed an error in the stratospheric ozone model; comparison with new ozone metrics show that model performance has generally improved; comparison with physical climate metrics shows little change. For E3SMv1 at least, we find that 20% errors in lower stratospheric ozone affect wave propagation, the tropopause, and the stability of the Antarctic stratospheric vortex. While these are readily detectable,
they seem to have much less impact on the fidelity of the climate simulation, or the climate sensitivity.

*Code availability.* The E3SM model is described in detail at https://e3sm.org/ and the source codes are available on GitHub: https://github.com/E3SM-Project/E3SM.

*Author contributions.* QT and MJP designed the experiments and the scope and structure of the manuscript. QT carried out the simulations and analyzed the data. QT prepared the manuscript with contributions from all co-authors.

*Acknowledgements.* This research was primarily supported by the Energy Exascale Earth System Model (E3SM) project, funded by the U.S. Department of Energy (DOE), Office of Science, Office of Biological and Environmental Research (BER) under the auspices of the U.S.
DOE by Lawrence Livermore National Laboratory under contract DE-AC52-07NA27344 and University of California, Irvine (UCI). This research was also partially supported by the Climate Model Development and Validation activity, funded by the Office of BER in the U.S. DOE Office of Science. UCI also acknowledges support from NASA Atmospheric Chemistry Modeling and Analysis Program grant 80NSSC20K1237. This research used resources of the National Energy Research Scientific Computing Center, a DOE Office of Science User Facility supported by the Office of Science of the U.S. DOE under Contract No. DE-AC02-05CH11231 and the BER Earth System
Modeling program's Compy computing cluster located at Pacific Northwest National Laboratory (PNNL). PNNL is operated by Battelle for the U.S. DOE under Contract DE-AC05-76RL01830. LLNL-JRNL-813913.



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

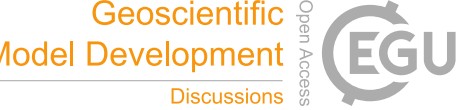

**Table 1: List of simulation configurations, periods, and brief purposes**

| Setting | Years | Purpose |
|---|---|---|
| O3v1 AMIP | 1870-2014 | Control run from the E3SMv1 DECK |
| O3v2 AMIP | 1990-2014 | O3v2 test run |
| UCI CTM | 1990-2017 | Same O3v2 but using ECMWF circulation and a 199 K PSC T threshold |
| O3v2 F2010 | 0001-0005 | Cess control experiment |
| O3v2 F2010+4K SST | 0001-0005 | Cess experiment with +4K SST |

**Table 2: List of evaluation data sets**

| Instrument | Years | Specifications | Reference |
|---|---|---|---|
| Aura OMI & MLS | 2005-2017 | 1º, 60ºS-60ºN, monthly zonal SCO | (Ziemke et al., 2019) |
| Aura MLS | 2005-2019 | 4º, 82ºS-82ºN, <216 hPa, monthly zonal O3 profile | (Schwartz et al., 2020) |
| Nimbus-7 TOMS, Aura OMI, Suomi NPP OMPS | 1979-2019 | Daily O3 hole area, SH minimum TCO | (https://ozonewatch.gsfc.nasa.gov, accessed on 29 May 2020) |

**Table 3: Net climate feedback parameter (λ, unit: W/m2/K) of different E3SM configurations**

| | O3v1 | O3v2 | O3v1 HR | O3v1 LRtunedHR |
|---|---|---|---|---|
| λ | -1.36 | -1.38 | -1.29 | -1.20 |





**Figure 1: Multi-year mean annual cycle of the zonal mean stratosphere-only column ozone (SCO, in Dobson Units). The SCO from (a) OMI+MLS observations are for years 2005-2017; that from (b) UCI CTM, years 2005-2017; that from E3SM (c) O3v1 and (d) O3v2, years 1995-2014 as forced by observed SSTs. Comparison with observed SCO is limited to 60°S-60°N limited by the better observational data. (e) The difference in SCO of O3v2 minus O3v1 for 90°S-90°N.**



**Figure 2: The standard deviation (in Dobson Units) of the zonal mean SCO monthly anomalies relative to the long-term average in Figure 1. The OMI+MLS observations are for years 2005-2017. The model results show long-term interannual variability and are from 2 different 13-year periods: E3SM – 1995-2007 (dash lines) and 2002-2014 (solid lines); UCI CTM – 1992-2004 (dash line) and 2005-2017 (solid line).**

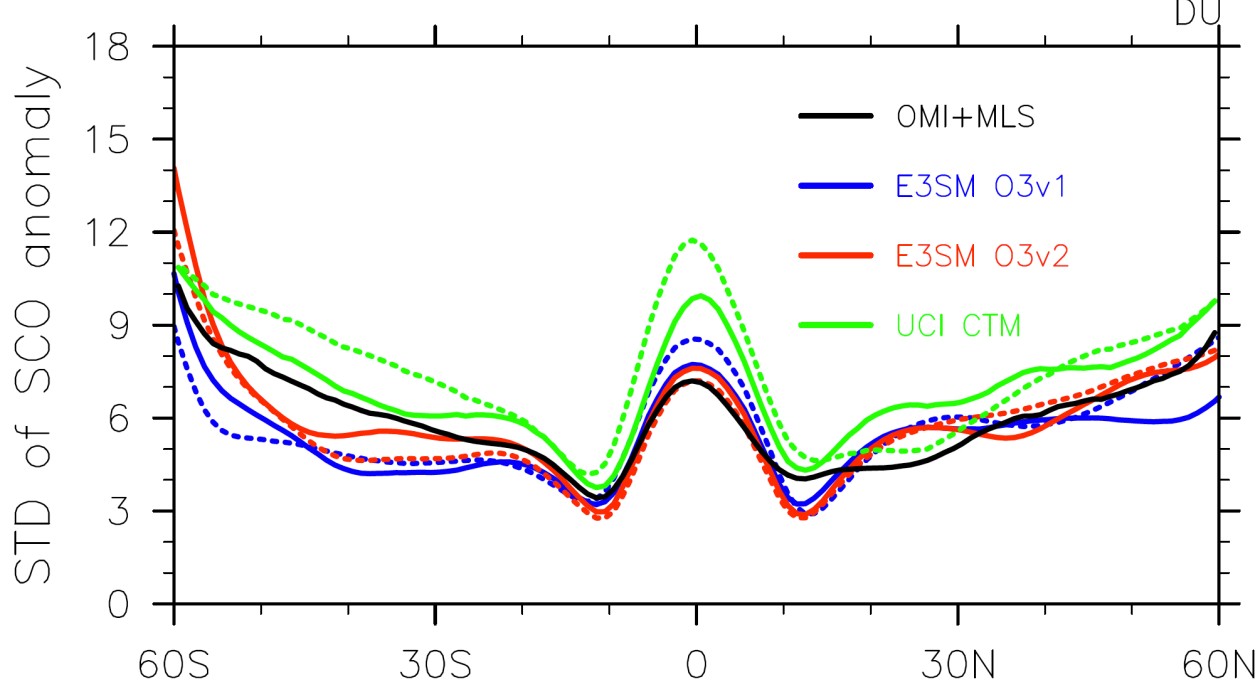





**Figure 3: Taylor diagrams of (a) area-weighted multi-year annual cycle of SCO (Figure 1) and area-weighted year-to-year standard deviation of the SCO monthly anomalies (Figure 2) with the OMI+MLS observations as the reference point (1,0); (b) area-weighted multi-year zonal mean stratospheric ozone abundances (Figure 4) relative to the MLS observations. Results are shown only for annual and June means, other months are similar; (c) area-weighted year-to-year standard deviation of the zonal mean stratospheric ozone abundances relative to the MLS observations; (d) daily Antarctic ozone hole diagnostics (Figure 5) relative to the NASA ozone watch data. Numbers 1 and 3 are for the daily mean time series, while numbers 2 and 4 represent the daily STD time series. On all Taylor diagrams, the model STDs are normalized by dividing the STDs of the reference (labeled with units).**



**Figure 4: Latitude by pressure plots of multi-year zonal mean stratospheric ozone abundances (in parts per million mole fraction, ppm). The 3 columns show annual mean, June, and October (left to right). The rows show MLS observations, E3SM O3v1, E3SM O3v2, and UCI CTM (top down).**



**Figure 5: (left) Time series of zonal mean SCO by latitude (unit: DU) showing Antarctic ozone hole over years 1995-2014 and (middle) daily evolution of the ozone hole from July 1 to December 31 as measure by area ($10^6$ km$^2$), and (right) minimum total column ozone (DU). Results are shown for models E3SM O3v1 (top row) and O3v2 (bottom row). Results for UCI CTM are not shown because daily diagnostics were not saved. Observations from the NASA ozone watch data for 1990-2019 are also shown in the right two columns. The lines indicate the multi-year mean (observations in black; models in blue), and shaded area covers ±1 standard deviation.**

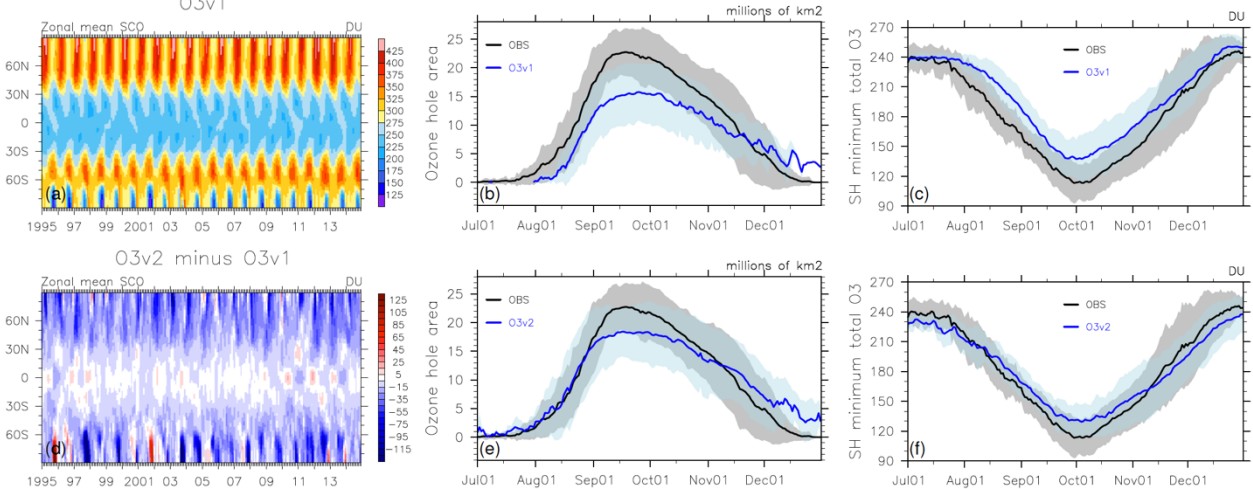

**Figure 6: Mean annual cycle of the stratosphere troposphere exchange (STE) ozone fluxes (unit: Tg O$_3$/yr) at NH (solid lines) and SH (dashed lines) for O3v2 (red) and UCI CTM (green). The lines indicate the climatology mean, and shaded area covers the mean ±1 STD.**

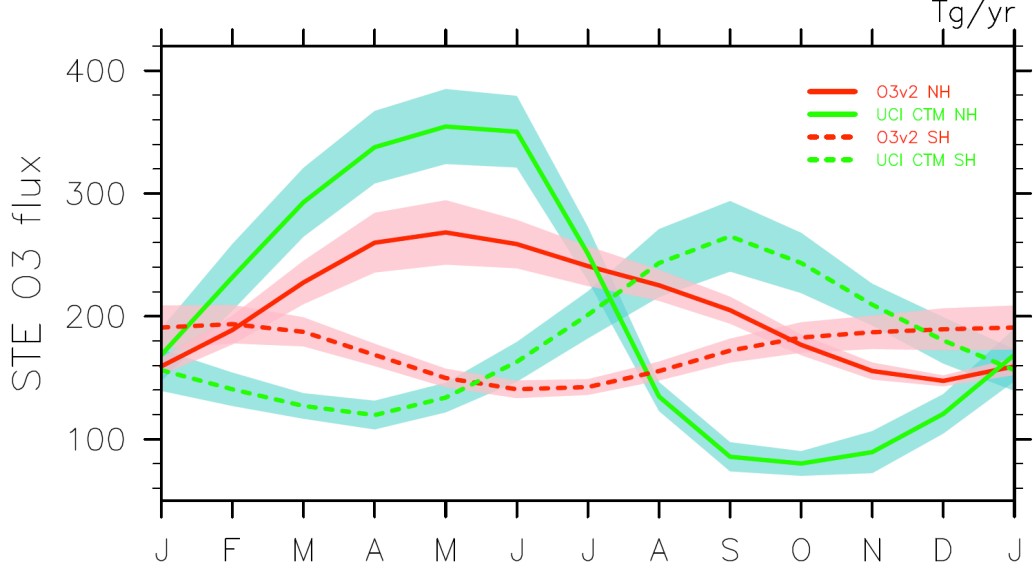



**Figure 7: Zonal mean SCO profiles from O3v1 (top) and O3v2 minus O3v1 (bottom) at the upper troposphere/lower stratosphere (UT/LS) region for annual, June, and October means.**

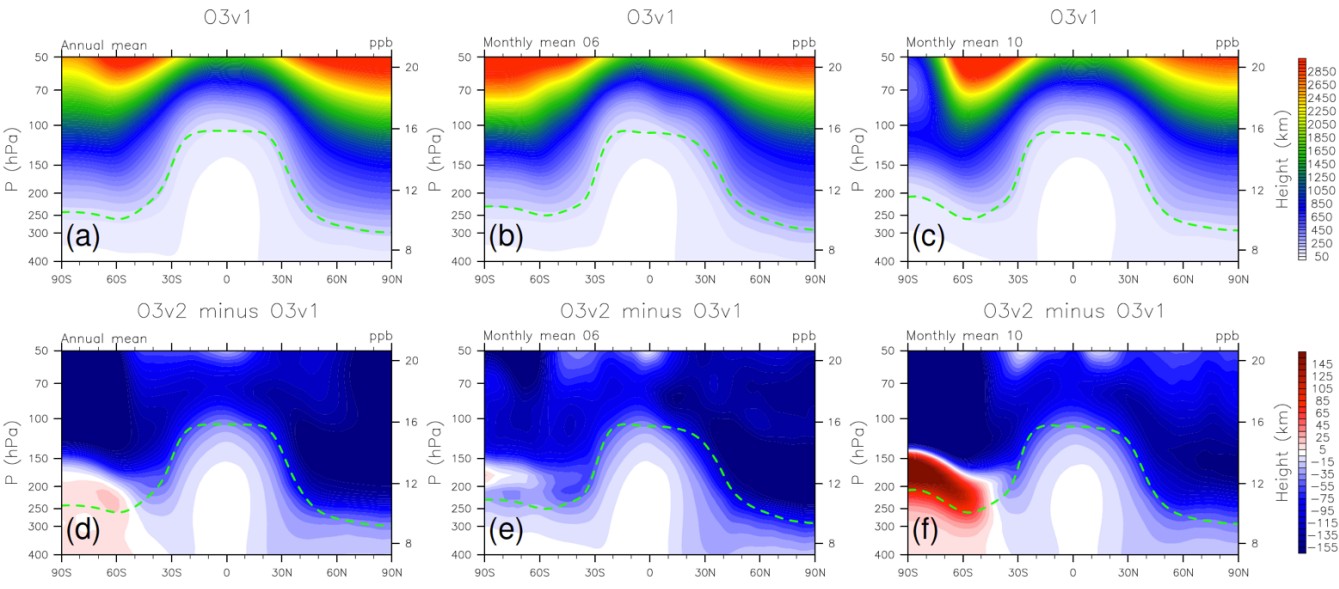

5    **Figure 8: Same as Fig. 7, but for total net heating.**

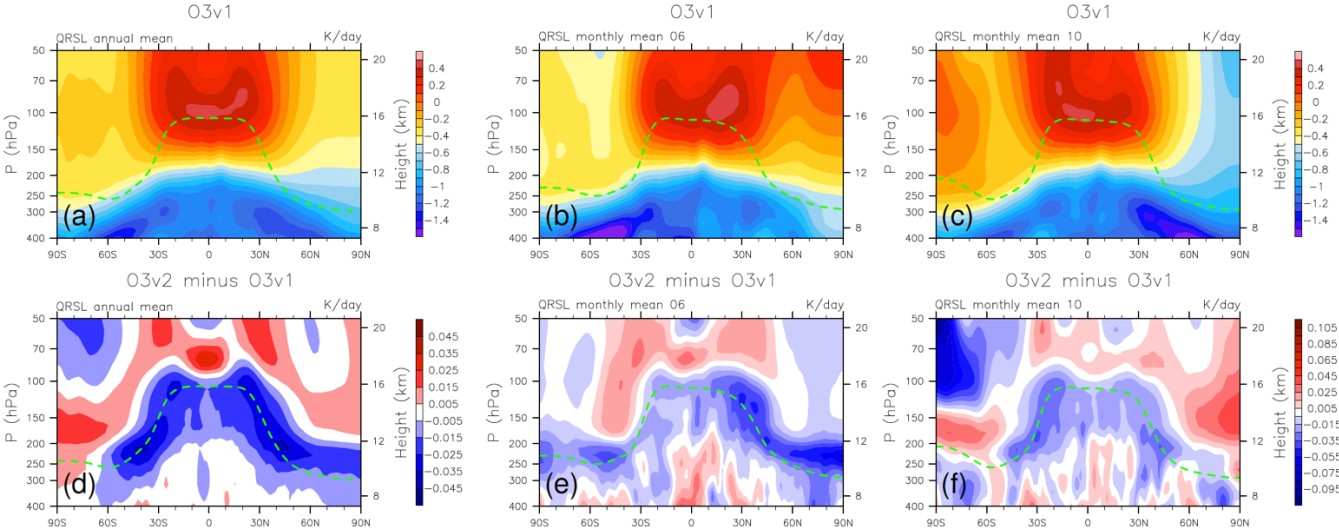



**Figure 9: Zonal mean tropopause changes between O3v2 and O3v1 for annual and monthly means.**






**Figure 10: Comparison of global uncentered RMSE (1981–2005) of an ensemble of 30 Coupled Model Intercomparison Project Phase 5 (CMIP5) models (box and whiskers showing 25th and 75th percentiles, minimum and maximum) with the two E3SM Atmospheric Model Intercomparison Project (AMIP) simulations (O3v1: blue dots; O3v2: red dots). Spatial RMSE against observations are computed for annual and seasonal averages with the PCMDI Metrics Package (Gleckler et al., 2016). Fields**

5 **shown include (a) top-of-atmosphere (TOA) net radiation, TOA (b) SW and (c) LW cloud radiative effects, (d) precipitation, (e) surface air temperature over land, (f) zonal wind stress over ocean, (g) 200- and (h) 850-hPa zonal wind, and (i) 500-hPa geopotential height. SW = shortwave; CRE = cloud radiative effects; LW = longwave; DJF = December–February; MAM = March–April; JJA = June–August; SON = September–November; RMSE = root-mean-square error.**



**Figure 11: Same as Fig. 10, but for 50S-90S.**





**Figure A1: Pressure by latitude mean differences between O3v2 without retuning the PSC T threshold and O3v1 in October of 5 weak ozone hole years (1998, 2001, 2007, 2011, 2013) for (a) O₃, (b) temperature, (c) zonal wind, (d) PSC ozone loss tendency, (e) buoyancy frequency squared (N²), and (f) E-P flux vector and divergence.**

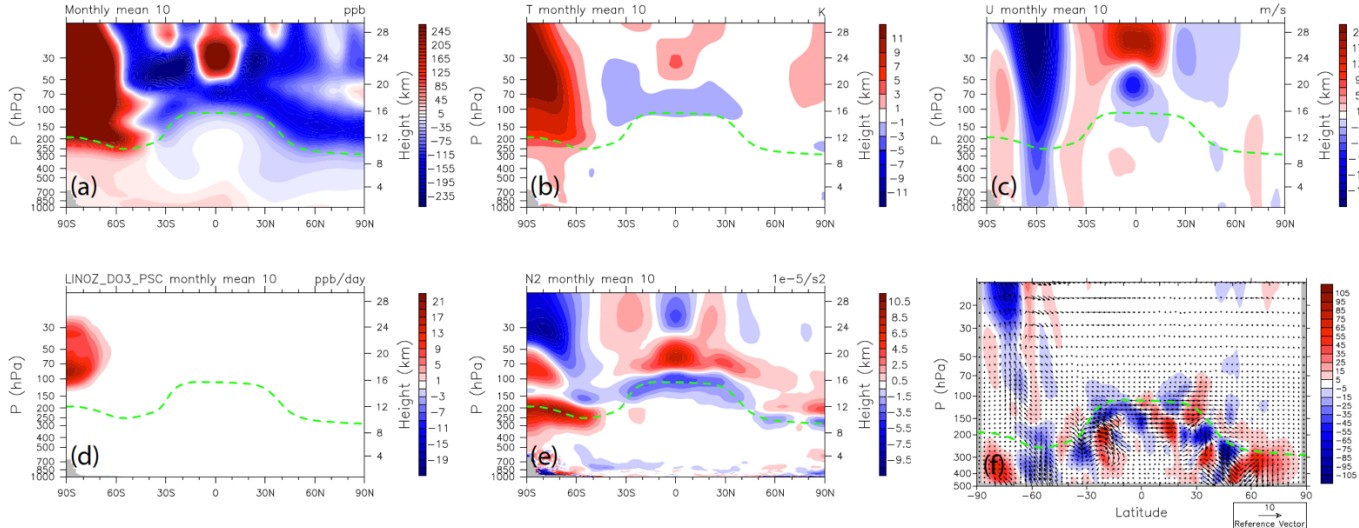

