# Peer review of "Evaluation of the interactive stratospheric ozone (O3v2 module) for the E3SM version 2 Earth System Model"

_Geoscientific Model Development, 2020_

## Referee Comment (RC1) · Anonymous Referee #1 · 19 Oct 2020

The Tang et al. manuscript presents a revised approach to parameterizing the distribution of ozone for use in the model radiation schemes. The approach is based on the widely used Linoz scheme in the stratosphere, but replaces the specified ozone previously used in the troposphere with the ozone tracer predicted by Linoz, subject to an imposed lower boundary condition of 30 ppb near the surface. The advantage of the revised approach is a more consistent specification of ozone around the tropopause, particularly in the case where the tropopause is higher than the climatological average implicit in the specified tropospheric ozone climatology. The two approaches are compared by implementing them in the US Department of Energy's Energy Exascale Earth System Model (E3SMv1) for AMIP-type simulations using specified sea-surface

temperatures for 1995 – 2014.

In general, the manuscript presents the results in a scientifically valid way and I have only minor comments. I would be a bit critical of the presentation of the material, however. The manuscript spends some time comparing various aspects of the ozone climatology that results from the two parameterizations installed in E3SMv1 against observations, here I am referring to the results shown in Figures 1 – 4. It is only a bit later that the impacts of the revised ozone parameterization on the model dynamics is discussed and the reader realizes that the differences discussed during the first part of the paper are not just the results of the revised ozone parameterization but also result from significant changes in the model dynamics. The authors could provide some overview of the situation earlier in the manuscript. I would also offer some criticism of the fact that there are significant differences in the ozone cross section shown in Figure 7, particularly in the lower stratosphere. While the problems with the original ozone parameterization in the vicinity of the tropopause are easy to imagine, there is no specific comparison against observations to show that the revised ozone parameterization produces a more correct ozone mixing ratio in this region.

Minor Corrections:

The title: I might suggest moving the word 'module' outside of the brackets so that it is 'stratospheric ozone (O3v2) module for...'

Page 1, Lines 21-23: The sentence 'As expected, SST forcing does not match the observed quasi-biennial oscillation...' could more clearly convey information. Having a free-running atmosphere forced only by observed SSTs does not match the observed QBO. It is not clear if there is a QBO spontaneously generated in E3SMv1 and the problem is that the model QBO is not in phase with the observed QBO, or whether there is not a QBO at all in E3SMv1. Page 6, Lines 4 – 8: The O3v1 control is described as being one of the three AMIP simulations forced with prescribed SSTs and sea-ice. The CMIP6 AMIP experiment that is part of the DECK is specified as running from

1979 – 2014 using observed SSTs and sea-ice, so the wording used here can be a bit confusing because AMIP refers to several runs of different length using specified SSTs and sea-ice. Perhaps 'AMIP-type' would be less confusing. The text also does not specifically mentioning whether the SSTs/sea-ice are from observations. The source of the SSTs/sea-ice should also be stated here. HadISSTs, I assume?

Page 6, Lines 14 – 17: Coming back to the use of the word AMIP to mean any run with specified SSTs/sea-ice, here the text refers to a pair of AMIP simulations where one of them uses SSTs increased by 4K. I understand what you mean by AMIP, but AMIP has a specific usage in CMIP and it is being used considerably more loosely here, particularly when SSTs are increased by 4K. I would suggest moving away from referring to all the runs as 'AMIP simulations'. Page 6, Line 17: Do you modify the sea-ice at all for the case where the SSTs are uniformly increased by 4K? This is not an objection to any particular treatment of sea-ice, just that it would be helpful to better understand how the simulation was set up.

Page 9, Line 12: For the STD/SCO shown in Figure 2, there are two time periods plotted up. Is the Taylor diagram in Figure 3 for the two period combined or only one of them?

Page 9, Lines 21 -23: On the improvement in the RMS error in the standard deviation of the SCO for O3v2, have you considered just internal variability? Both O3v1 and O3v2 are from free-running simulations of the E3SMv1 and they do show some regions of significant difference between the two period sampled.

Page 11, Lines 1 – 12: It is interesting that for both versions of E3SM the RMS for the annual average standard deviation (Figure 3c) is larger than for any of the individual months. It is a bit more difficult to see, but the annual average for UCI is within the cloud of points formed by the individual months. Do you have any ideas for why the RMS of the annual average standard deviation for E3SM would be larger than for any of the months?

Page 11, Lines 23 - 27: Figure 5 shows the time evolution of zonal average ozone for O3v1 and the difference O3v2 – O3v1. There is the differences in Northern hemisphere ozone, with O3v1 having much larger columns and can be more easily seen in Figure 1, but given the year-to-year variability in dynamics a straight year-to-year difference plot is significantly affected by the dynamical variability. I would suggest the authors replace the O3v2 – O3v1 difference with a plot of the O3v2 column as this would more clearly show the ozone hole behaviour in O3v2.

Page 14, Lines 1 – 4: The cross-section of ozone changes between O3v1 and O3v2 shown in Figure 7 show absolute differences which makes it difficult to assess the correctness of the statement 'with O3v2 having about 20% less ozone in the lower stratosphere, but hardly any change in the troposphere...'. I would ask the authors to consider adding cross-section plots of the percentage differences - if not as a replacement for the current panels in Figure 7 then as an additional figure in the appendix?

Page 34, Caption to Figure A1. Here a list of five years with weak Antarctic ozone depletion is given for which the differences between O3v1 and O3v2 are calculated. If both simulations are free-running and have the ozone interactive with the model, shouldn't the years with weak ozone be different between the two simulations?

---

## Short Comment (SC1) · 27 Oct 2020

Dear authors,

the GMD manuscript type "evaluation paper" is mainly for models, which descriptions have been already per-reviewed published elsewhere. But in your case, if I understand correctly, you did develop new code for this publication: namely O3v2. Therefore all code availability requirements of the model description paper type apply here: In particular your discussion paper does not meet the below cited requirement:

- Code must be published on a persistent public archive with a unique identifier for

[Figure]

the exact model version described in the paper or uploaded to the supplement, unless this is impossible for reasons beyond the control of authors. All papers must include a section, at the end of the paper, entitled "Code availability". Here, either instructions for obtaining the code, or the reasons why the code is not available should be clearly stated. It is preferred for the code to be uploaded as a supplement or to be made available at a data repository with an associated DOI (digital object identifier) for the exact model version described in the paper. Alternatively, for established models, there may be an existing means of accessing the code through a particular system. In this case, there must exist a means of permanently accessing the precise model version described in the paper. In some cases, authors may prefer to put models on their own website, or to act as a point of contact for obtaining the code. Given the impermanence of websites and email addresses, this is not encouraged, and authors should consider improving the availability with a more permanent arrangement. Making code available through personal websites or via email contact to the authors is not sufficient. After the paper is accepted the model archive should be updated to include a link to the GMD paper.

Please provide the exact code version including your new code developments in a permanent archive (i.e. with DOI or other persistent identifier). The newly developed code must be available to the referees during the discussion phase. For projects in GitHub a DOI for a released code version can easily be created using Zenodo, see https://guides.github.com/activities/citable-code/ for details.

Yours, Astrid Kerkweg

---

## Author Comment (AC1) · 27 Oct 2020

Thanks for the comment.

The exact code (including the newly developed code) used in this paper is publicly available at https://github.com/E3SM-Project/E3SM/tree/tangq/maint-1.0/O3v2_PSC with a hash of v1.0.0-193-g17aa44d. We will revise our manuscript to meet the GMD requirements like we did for our previous GMD papers.

---

## Referee Comment (RC2) · Anonymous Referee #2 · 2 Nov 2020

GENERAL COMMENTS

This paper reports on the new ozone chemistry module developed and implemented in the U.S. Department of Energy's Energy Exascale Earth System Model version 1 (E3SMv1). The paper is well suited for publication in GMD and I believe will be of interest to a wide range of readers of GMD. While it may look like I have suggested many changes below, none of them are substantive and I expect that the authors can work through these and implement them (or not) within a couple of weeks. As such, I suggest that this paper can be published with minor corrections.

SPECIFIC COMMENTS
Page 1, Line 9: Why 'feedbacks'? Often these are just one-way processes and not feedbacks in the strict sense of the word i.e. A affects B and then B either affects A or affects things that affect A.

Page 1, Line 16: Presumably stratosphere-troposphere exchange of ozone was implemented in E3SMv1 but was simply not tracked or diagnosed?

Page 1, Line 18: Satellite observations of what exactly? And what variables are you comparing here between E3SMv1 and UCI CTM? Just ozone or also other variables?

Page 1, Line 21-23: I found this sentence very confusing. I associate the QBO with the stratosphere and here you are talking about 'SST forcing does not match the observed quasi-biennial oscillation' and then 'mostly matched with the UCI CTM'. What does it mean for the QBO to be 'matched with' the UCI CTM? Do you mean that the UCI CTM simulates the QBO in stratospheric column ozone well?

Page 2, Lines 2-4: I see a rather large disconnect between the first and second sentences of the Introduction. I agree with the first sentence but when I think about climate models needing to represent GHG concentration distributions correctly, I think primarily about CO2, N2O, and CH4. My first thought is not atmospheric ozone. A better formulation of the first sentence would be 'Accurate simulation of past climate evolution and projections of future climate depend, rather weakly, on correct representation of atmospheric ozone'. But that is not a very motivating start to the paper. I would suggest rewriting the first sentence so that it better motivates why getting ozone right in climate models matters.

Page 2, line 14: Is it worth defining what 'full' means in this context? I don't know but maybe you should think about it.

Page 2, line 15: Is this true? My recollection, though I may be wrong, is that about 50% of the CMIP5 models had interactive ozone.

Page 2, line 28: It wasn't quite clear to me what you meant by 'greenhouse ozone-

depleting gases'. I guess you mean the CFCs and HCFCs? Gases with a non-zero ODP *and* non-zero GWP? It is an unfamiliar term (to me) and so maybe you want to consider using something better known.

Page 3, line 5: I think you should be more specific and say which 'other model'.

Page 3, line 8: It wasn't clear to me what you meant by 'these errors were not symmetrical'. Presumably the climatology overwrite would also place low tropospheric concentrations of ozone into the stratosphere? But what do you mean by 'these errors were not symmetrical'?

Page 4, line 10: Can you please cite a few papers that support the assertion that 'This model has proven robust and reasonably accurate'.

Page 4, line 13: I guess then that what you really want is a tropopause-indexed ozone climatology e.g. Sofieva, V.F.; Tamminen, J.; Kyrölä, E.; Mielonen, T.; Veefkind, P.; Hassler, B. and Bodeker, G.E., A novel tropopause-related climatology of ozone profiles, Atmospheric Chemistry and Physics, doi:10.5194/acp-14-283-2014, 2014?

Page 5, line 6: 'lower boundary sink' of ozone presumably?

Page 5, line 11: Again it is not clear to me what you mean by 'set to match the observed Antarctic ozone'? Do you mean that tunable parameters in the O3v1 module were set so that simulations of stratospheric ozone using this module would replicate the characteristics of the observed Antarctic ozone hole? If that is what you mean, perhaps that's what you should write.

Page 6, line 12: Is it driven by forecast winds or reanalysis winds?

Page 6, line 13: I know what you mean by 'time-specific observations' but other readers may not. Perhaps better to say 'the true state of the atmosphere rather than a state with the same climate but different weather as would be the case with E3SM'.

Page 7, line 11: Have the MERRA TCO data been validated? e.g. do you see a clear

discontinuity when you go from measured values to MERRA-filled values in any daily TCO field?

Page 7, line 13: The minimum TCO over what geographic domain?

Page 7, line 21: What is 'milli-cm-Amagats'? I have never seen that before. Wouldn't "1DU = 2.69 x 10^16 molecules/cm2)" make more sense to more people?

Page 8, line 6: Couldn't you also patch the SCO fields with MERRA ozone data or are the MERRA ozone data not vertically resolved?

Page 8, line 12: I would refer to these as 'biases' rather than 'errors'.

Page 8, line 12: What, exactly, is 'excellent in the tropics but too great at high latitudes'?

Page 8, line 14: But is still biased high right?

Page 8, line 20: By 'STD' I assume you mean standard deviation? I think you need to state that more clearly.

Page 8, line 26: In what way is this 'peak QBO-like variability'. You are not showing anything with a quasi-biennial oscillation time scale.

Page 8, line 28: It is not clear to me at all that 'STD/SCO provides a second test of the overall stratospheric circulation'. I think that you need to demonstrate that far more robustly. It is certainly not self-evident.

Page 9, line 7: Seeing the phrase 'is probably adequate' in a paper does not fill me with confidence. Can't you do the statistical test and make a definitive statement?

Page 9, lines 7-8: It wasn't clear to me what you meant by 'The jump in the longterm UCI CTM STD'? Do you mean the change in standard deviation between the two periods for which the UCI CTM standard deviations were calculated?

Page 9, line 8-9: I have never known a switch in satellite data to cause changes in the wind-driven circulation! That would be very impressive. I think that you need to be

much clearer in saying what you mean.

Page 9, lines 18-19: I don't understand the sentence 'The UCI CTM scores slightly better because of the high-latitude SCO' at all. First better than what? And second 'because of the high-latitude SCO' is not an explanation for anything.

Page 9, line 25: Does not always perform better than what?

Figure 4: I think it would be worth stating in the figure caption over what period these climatologies were calculated.

Page 10, lines 9-10: I don't think the sentence 'This metric has been a standard test for 2D and 3D stratospheric chemistry models for decades' is necessary. Let's say it had only ever been used once before. Would that make your analysis any less appropriate?

Page 10, line 10: While I could take a guess, it wasn't entirely clear to me what you meant by 'The model goal'. I think you should describe what you mean more clearly so that the reader doesn't have to guess.

Page 10, line 13: In what way are the seasonal upward shifts in the contours in the winter 'odd'? To me, they look entirely as you would expect.

Page 11, line 11: I have no idea what you mean by 'This metric is a tough one'? Tough like Sylvester Stallone or tough like Arnold Schwarzenegger? I was also confused by 'but we will need to add some other models to see how well it works outside of Linoz chemistry'. OK then go and add more models if that's what you need.

Page 11, line 17: This is somewhat true. Equivalent Effective Antarctic Stratospheric Chlorine increased quite a bit from 1990 to 2000 and then decreased more slowly thereafter.

Figure 5: I am surprised that you are using minimum TCO as a metric when Müller, R.; Grooß, J.-U.; Lemmen, C.; Heinze, D.; Dameris, M. and Bodeker, G.E., Simple measures of ozone depletion in the polar stratosphere, Atmospheric Chemistry and

Physics, 251-264, 8, 2008, warned against using it.

Page 11, line 22: Sorry by 'ozone column' do you now mean SCO or TCO?

Page 12, lines 6-7: So why wasn't O3v1 tuned with a better PSC temperature threshold?

Page 12, line 10: I suspect you mean Figure 3d here?

Page 12, lines 10-13: It is not clear to me what you mean by the 'dynamical conditions'...'remain relatively isolated from the ozone hole chemistry'? Dynamical conditions play a huge role in the efficacy of ozone depletion chemistry in the Antarctic stratosphere. That's what accounts for all of the interannual variability in Antarctic ozone depletion.

Page 12, lines 13-15: I don't understand the purpose, meaning, or relevance of these last two sentences. Unless you have compelling reasons not to, I would suggest just deleting them.

Page 12, line 22: By 'STE flux' do you mean the flux in general (i.e. kg/m^2/sec) or do you mean the ozone flux specifically?

Page 13, lines 3-4: There is no place for a sentence like this is a paper. Either you did collect 'enough different models with enough similar results' to build a Taylor diagram or you didn't. So which is it? Otherwise what are you hoping for the reader to conclude from this sentence? It seems like speculation with no purpose.

Page 13, line 6: I find this sentence very confusing. What, exactly, is set to the lowest four layers? The tropospheric ozone loss? But why would that be quantitatively equivalent to the STE ozone flux? Maybe I am misunderstanding something here? But if I am, it is possible that other readers would too. I think that you need to explain yourself much more clearly here.

Page 13, line 7: Do you really mean 'averaged over latitude and month'? So you have

a monthly mean for every longitude? I wouldn't understand why you would do that.

Page 13, line 10: But how are you getting zonal means when you averaged over latitude (I am assuming you meant averaged over all latitudes)?

Page 13, line 17-18: Replace 'peaks in May and bottoms in Dec' with 'maximizes in May and minimizes in December'.

Page 13, line 18: Regarding 'the peak extends to Jun'. Here and throughout, there is no need to use abbreviations for months in the manuscript text. That extra 'e' isn't going to blow out your publications budget.

Page 14, line 4: Are you going to be examining the changes between O3v1 and O3v2 in greater detail (i.e. digging into how the coding of O3v1 and O3v2 differs) or are you going to be examining how changes from O3v1 to O3v2 affect the distribution of ozone in the UT/LS etc.? My primary complaint about this paper is that your are being too vague in your writing and it is often not clear exactly what you mean.

Page 17, line 21: I don't understand what you mean by 'defined proportional to the reverse of lambda'. Do you mean the inverse of lambda?

Page 18, line 10: Delete the sentence 'Running these metrics with O3v1, O3v2, and the UCI CTM was informative.' If this wasn't the case you shouldn't have written the paper so to some extent it is self-evident.

Page 18, line 14: In what way is the temperature threshold for PSC formation 'delicate'?

Page 19, line 10: I would suggest informal editorial comments such as 'So we must accept good fortune:' should have no place in a paper. Unless of course you can cite a paper, perhaps from the humanities, that supports the assertion that 'we must accept good fortune'.

Page 19, line 15: I don't like to see phrases such as 'they seem to have much less impact on the fidelity' in a paper. Either it does impact the fidelity or it doesn't. Do a

test so that you can state categorically which it is.

GRAMMAR AND TYPOGRAPHICAL ERRORS

I understand that the author's first language may not be English. It is not for me to say, but perhaps the second author could wordsmith the paper? To meet the standard of writing required for this journal, the quality of the writing needs to be improved, unless the journal employs a copy editor to do so. The list of corrections I have listed below is not complete and in some cases may only reflect my own writing style. They should be taken as suggestions.

Page 1, line 20: I think it would be clearer to say 'reduced bias' rather than 'shows improved mean bias'.

Page 2, line 15: Replace 'adopt mean climatological distribution' with 'adopt a mean climatological distribution of ozone'.

Page 3, line 1: Replace 'CTM' with 'CTMs'.

Page 3, line 16: Should this be 'E3SMv1' rather than just 'E3SM'? Likewise on line 23. Please ensure consistency in nomenclature throughout.

Page 4, line 23 and elsewhere: I would suggest you use the word 'shortcomings' rather than 'problems' in this context.

Page 4, line 24: Replace 'it is removed' with 'where it is removed' otherwise the grammar is wrong.

Page 5, line 19: Do you mean 'forced only with observed SSTs'?

Page 5, line 22: Delete 'check'.

Page 5, line 26: I would suggest shifting this sentence starting 'It would be interesting...' to the conclusions section where you might provide an 'outlook' of how this research could be extended.

Page 6, line 6: You have already defined the SST acronym earlier. You don't need to define it again.

Page 6, line 10: Replace 'through the end' with 'through to the end'.

Page 6, line 15: Replace 'are carried' with 'is carried'.

Page 7, line 1: I would suggest 'tropospheric ozone column' rather than 'tropospheric column ozone'.

Page 7, line 8: Replace 'on the NASA's' with 'on NASA's'.

Page 8, lines 21-22: This sentence is grammatically incorrect.

Page 9, line 15: Replace 'suggesting well-captured annual' with 'suggesting a well-simulated annual'.

Page 10, line 3: Replace 'as diagnostic of chemistry' with 'as a diagnostic of chemistry'.

Page 11, line 26: 'frequent minimal values like 1995 and 2001' is worded poorly and should be rephrased.

Page 14, line 12: Replace 'at the lower stratosphere' with 'in the lower stratosphere'.

Page 14, line 22: Replace 'tropopause to the stratosphere' with 'troposphere to the stratosphere'.

Page 16, line 21: Replace 'thermo-dynamical' with 'thermodynamic'.

Page 17, line 3: Replace 'associated to' with 'associated with'.
* * *

---

## Author Comment (AC2) · 16 Dec 2020

We thank both reviewers for their useful and constructive comments to help improve the quality of the paper. We revised our manuscripts based on the suggestions. Below are the detailed responses (reviewer comments in blue and our replies in black).

**Anonymous Referee #1**

The Tang et al. manuscript presents a revised approach to parameterizing the distribution of ozone for use in the model radiation schemes. The approach is based on the widely used Linoz scheme in the stratosphere, but replaces the specified ozone previously used in the troposphere with the ozone tracer predicted by Linoz, subject to an imposed lower boundary condition of 30 ppb near the surface. The advantage of the revised approach is a more consistent specification of ozone around the tropopause, particularly in the case where the tropopause is higher than the climatological average implicit in the specified tropospheric ozone climatology. The two approaches are compared by implementing them in the US Department of Energy's Energy Exascale Earth System Model (E3SMv1) for AMIP-type simulations using specified sea-surface temperatures for 1995 – 2014.

In general, the manuscript presents the results in a scientifically valid way and I have only minor comments. I would be a bit critical of the presentation of the material, however. The manuscript spends some time comparing various aspects of the ozone climatology that results from the two parameterizations installed in E3SMv1 against observations, here I am referring to the results shown in Figures 1 – 4. It is only a bit later that the impacts of the revised ozone parameterization on the model dynamics is discussed and the reader realizes that the differences discussed during the first part of the paper are not just the results of the revised ozone parameterization but also result from significant changes in the model dynamics. The authors could provide some overview of the situation earlier in the manuscript. I would also offer some criticism of the fact that there are significant differences in the ozone cross section shown in Figure 7, particularly in the lower stratosphere. While the problems with the original ozone parameterization in the vicinity of the tropopause are easy to imagine, there is no specific comparison against observations to show that the revised ozone parameterization produces a more correct ozone mixing ratio in this region.

We agree that it's helpful to foreshadow the polar dynamics changes earlier, so that the readers have a better overview of the impacts of the new ozone parameterization. We added "We should mention that besides different ozone the new O3v2 parameterization causes unexpected changes to the dynamics over the southern polar region. We will discuss these dynamics changes in Section 4." to the end of the first paragraph of Section 3 (Page 8, Line 2).

While we don't have specific comparison against observations for the ozone near the tropopause (which is a very difficult comparison given the very sharp increase in ozone above the tropopause), our Taylor diagram (Figure 3a) of the stratospheric column ozone (SCO)

against the satellite observations shows O3v2 has a better SCO, which is clearly caused by improvements in the lowermost stratosphere.

Minor Corrections:
The title: I might suggest moving the word 'module' outside of the brackets so that it is 'stratospheric ozone (O3v2) module for...'

Thanks. Revised.

Page 1, Lines 21-23: The sentence 'As expected, SST forcing does not match the observed quasi-biennial oscillation...' could more clearly convey information. Having a free-running atmosphere forced only by observed SSTs does not match the observed QBO. It is not clear if there is a QBO spontaneously generated in E3SMv1 and the problem is that the model QBO is not in phase with the observed QBO, or whether there is not a QBO at all in E3SMv1.

The SST-forced E3SMv1 can simulate a QBO, but cannot match the observation (too frequent and too strong) as shown by Richter et al. (2019). We slightly modified the text here to better convey this message.

Richter, J. H., Chen, C.-C., Tang, Q., Xie, S., & Rasch, P. J. (2019). Improved simulation of the QBO in E3SMv1. Journal of Advances in Modeling Earth Systems, 11, https://doi.org/10.1029/2019MS001763.

Page 6, Lines 4 – 8: The O3v1 control is described as being one of the three AMIP simulations forced with prescribed SSTs and sea-ice. The CMIP6 AMIP experiment that is part of the DECK is specified as running from 1979 – 2014 using observed SSTs and sea-ice, so the wording used here can be a bit confusing because AMIP refers to several runs of different length using specified SSTs and sea-ice. Perhaps 'AMIP-type' would be less confusing. The text also does not specifically mentioning whether the SSTs/sea-ice are from observations. The source of the SSTs/sea-ice should also be stated here. HadISSTs, I assume?

Good point, we changed to "AMIP-type". We also added the source of SSTs and sea ice: v1.1.3 of the Program for Climate Model Diagnosis and Intercomparison (PCMDI) data (Durack and Taylor, 2017; Taylor et al., 2000), which merges SST based on UK MetOffice HadISST and NCEP OI2.

Durack, P. J., & Taylor, K. E. (2017). PCMDI AMIP SST and sea-ice boundary conditions version 1.1.3. https://doi.org/10.22033/ESGF/input4MIPs.1735

Taylor, K. E., Williamson, D., & Zwiers, F. (2000). The sea surface temperature and sea ice concentration boundary conditions for AMIP II simulations (PCMDI Report 60). Livermore, CA: Program for Climate Model Diagnosis and Intercomparison, Lawrence Livermore National Laboratory. https://pcmdi.llnl.gov/report/pdf/60.pdf

Page 6, Lines 14 – 17: Coming back to the use of the word AMIP to mean any run
with specified SSTs/sea-ice, here the text refers to a pair of AMIP simulations where
one of them uses SSTs increased by 4K. I understand what you mean by AMIP, but
AMIP has a specific usage in CMIP and it is being used considerably more loosely
here, particularly when SSTs are increased by 4K. I would suggest moving away from
referring to all the runs as 'AMIP simulations'.

Yes, we agree, and have changed to use "AMIP-type simulations" throughout the manuscript.

Page 6, Line 17: Do you modify the
sea-ice at all for the case where the SSTs are uniformly increased by 4K? This is not
an objection to any particular treatment of sea-ice, just that it would be helpful to better
understand how the simulation was set up.

We only modified the SSTs (no sea-ice changes) to be consistent with the Cess experiments in
Caldwell et al. (2019) to facilitate the comparison with their results.  The text is modified.

Page 9, Line 12: For the STD/SCO shown in Figure 2, there are two time periods
plotted up. Is the Taylor diagram in Figure 3 for the two period combined or only one of
them?

We used the combined period of 1995-2014 for the EAM versions, whereas 2005-2017 for the
UCI CTM to match the period of satellite observations. We clarified this in both the text and the
figure caption.

Page 9, Lines 21 -23: On the improvement in the RMS error in the standard deviation of
the SCO for O3v2, have you considered just internal variability? Both O3v1 and O3v2
are from free-running simulations of the E3SMv1 and they do show some regions of
significant difference between the two period sampled.

Internal variability can lead to some differences in the O3v1-O3v2 RMS error of the SCO STD,
but it is not the dominant factor. The E3SM O3v1 result is from one of the three ensemble
members (Golaz et al., 2019). We looked at the results from the other two members. The
internal variability inferred by the differences in the ensemble cannot explain the differences
between O3v1 and O3v2 as shown here.

Page 11, Lines 1 – 12: It is interesting that for both versions of E3SM the RMS for the
annual average standard deviation (Figure 3c) is larger than for any of the individual
months. It is a bit more difficult to see, but the annual average for UCI is within the
cloud of points formed by the individual months. Do you have any ideas for why the
RMS of the annual average standard deviation for E3SM would be larger than for any
of the months?

We too suspected the possibility of some errors in the calculation when we first saw this result and went back to check the scripts. However, the same script was used to process the E3SM and UCI data and we concluded that the calculation was correct. We suspect that the larger E3SM RMS in the annual average standard deviation than in individual months is due to the non-linear calculations in the standard deviation and the RMS. We hesitate to put forward an answer without substantial new runs and diagnostics.

Page 11, Lines 23 - 27: Figure 5 shows the time evolution of zonal average ozone for O3v1 and the difference O3v2 – O3v1. There is the differences in Northern hemisphere ozone, with O3v1 having much larger columns and can be more easily seen in Figure 1, but given the year-to-year variability in dynamics a straight year-to-year difference plot is significantly affected by the dynamical variability. I would suggest the authors replace the O3v2 – O3v1 difference with a plot of the O3v2 column as this would more clearly show the ozone hole behaviour in O3v2.

The main point of Figures 5a and 5d is to demonstrate the O3v2-O3v1 differences in their simulated ozone hole as reflected by the zonal mean SCO time series. We feel the difference plot illustrates this point better than the O3v2 SCO plot.

Page 14, Lines 1 – 4: The cross-section of ozone changes between O3v1 and O3v2 shown in Figure 7 show absolute differences which makes it difficult to assess the correctness of the statement 'with O3v2 having about 20% less ozone in the lower stratosphere, but hardly any change in the troposphere...'. I would ask the authors to consider adding cross-section plots of the percentage differences - if not as a replacement for the current panels in Figure 7 then as an additional figure in the appendix?

The comparison of tropospheric ozone here is totally artificial: in O3v1 the values are simply the climatology from the input4MIPS Ozone data set v1.0 (Hegglin et al., 2016); while in O3v2 they reflect the gradient from tropopause to surface boundary condition. The reason to compare them and find them similar is important in terms of the tropospheric radiative forcing from ozone, since we wanted to avoid different climates from this cause. Fortunately, the largest heating differences are in the lowermost stratosphere as we anticipated. We added the cross-section plot of ozone relative changes as Fig. A1 in the appendix. The original Fig. A1 changed to Fig. A2.

Page 34, Caption to Figure A1. Here a list of five years with weak Antarctic ozone depletion is given for which the differences between O3v1 and O3v2 are calculated. If both simulations are free-running and have the ozone interactive with the model, shouldn't the years with weak ozone be different between the two simulations?

That's another way to composite the weak Antarctic ozone hole sample but will inevitably introduce the differences due to the fact that different SSTs and sea-ice forcings are used for different years. We opted to exclude the impact from different SSTs and sea-ice forcings to the ozone hole. We believe our choice is reasonable for the goal of the analysis here.

**Anonymous Referee #2**

GENERAL COMMENTS

This paper reports on the new ozone chemistry module developed and implemented in the U.S. Department of Energy's Energy Exascale Earth System Model version 1 (E3SMv1). The paper is well suited for publication in GMD and I believe will be of interest to a wide range of readers of GMD. While it may look like I have suggested many changes below, none of them are substantive and I expect that the authors can work through these and implement them (or not) within a couple of weeks. As such, I suggest that this paper can be published with minor corrections.

SPECIFIC COMMENTS

Page 1, Line 9: Why 'feedbacks'? Often these are just one-way processes and not feedbacks in the strict sense of the word i.e. A affects B and then B either affects A or affects things that affect A.

Agreed. We changed it to "reactions".

Page 1, Line 16: Presumably stratosphere-troposphere exchange of ozone was implemented in E3SMv1 but was simply not tracked or diagnosed?

E3SMv1 does full tracer transport of ozone, but it does not diagnose the flux across the 3D troposphere, which it would have to do every time step.  Defining the tropopause in 3D is difficult enough, and the only model we know whose tracer transport has been recoded to do this is the UCI CTM (e.g., Hsu and Prather, 2014, *Is the vertical residual velocity a good proxy for stratosphere-troposphere exchange of ozone?*  GRL, 41, doi:10.1029/2014GL061994).  It can be diagnosed less directly (i.e., not at the tropopause) as in some of the chemistry MIPs with tracer O3strat that has only chemical sinks in the troposphere.  The sum of these losses is the mean STE.  With O3v1, however, the chemistry arbitrarily resets the tropospheric O3 value to a climatology and this occurs even in the lower stratosphere when the tropopause is folded or mis-diagnosed due to the 2D definition of the tropopause in the E3SMv1.   So the problem is with the unphysical representation of O3 in O3v1.

Page 1, Line 18: Satellite observations of what exactly? And what variables are you comparing here between E3SMv1 and UCI CTM? Just ozone or also other variables?

Yes, the sentence is confusing, we broke the sentence to clarify that we used the ozone data from satellite observations, but also compared the STE ozone flux between E3SMv1 and UCI CTM.

Page 1, Line 21-23: I found this sentence very confusing. I associate the QBO with the stratosphere and here you are talking about 'SST forcing does not match the observed quasi-biennial oscillation' and then 'mostly matched with the UCI CTM'. What does it mean for the QBO to be 'matched with' the UCI CTM? Do you mean that the UCI CTM simulates the QBO in stratospheric column ozone well?

This sentence has been revised to
"As expected, SST-forced E3SMv1 simulations cannot synchronize with observed quasi-biennial oscillations (QBO), but they do show the typical QBO-pattern seen in column ozone."
The original was too complex and needed to focus on the new model.  The UCI CTM synchronizes and matches most of the observed record of column ozone, that is shown later in the paper.  Here we want to note that the E3SMv1 produces a composite QBO pattern like that observed.

Page 2, Lines 2-4: I see a rather large disconnect between the first and second sentences of the Introduction. I agree with the first sentence but when I think about climate models needing to represent GHG concentration distributions correctly, I think primarily about CO2, N2O, and CH4. My first thought is not atmospheric ozone. A better formulation of the first sentence would be 'Accurate simulation of past climate evolution and projections of future climate depend, rather weakly, on correct representation of atmospheric ozone'. But that is not a very motivating start to the paper. I would suggest rewriting the first sentence so that it better motivates why getting ozone right in climate models matters.

We disagree with the reviewer here: ozone is an important GHG, contributing 0.4 W/m2 to current warming.  We have revised the first two sentences to make that clear and give a better transition.
"Accurate simulation of past climate evolution and projections of future climate rely on correct representation of the greenhouse gases including ozone.  Simulating climate change driven by ozone is challenging for chemistry-transport modelling because ozone has two chemically distinct regions (stratosphere versus troposphere) with a very sharp interface at the tropopause. "

Page 2, line 14: Is it worth defining what 'full' means in this context? I don't know but maybe you should think about it.

Agreed, we have tried to better explain:
"Running a detailed atmospheric chemistry model for ozone, including both stratospheric and tropospheric chemical regimes, within a climate model is costly, often prohibitively, and thus most climate …"

Page 2, line 15: Is this true? My recollection, though I may be wrong, is that about 50% of the CMIP5 models had interactive ozone.

Eyring et al., 2013 Table 1 shows a nice summary of how ozone chemistry is treated in CMIP5 models. In the total of 46 models, 28 prescribed ozone, 9 used semi-offline chemistry, and only 9 included interactive chemistry. As much as we would like it, most CMIP5 climate models did not have interactive ozone.

Eyring, V., et al. (2013), Long-term ozone changes and associated climate impacts in CMIP5 simulations, *J. Geophys. Res. Atmos.*, 118, 5029– 5060, doi:10.1002/jgrd.50316.

Page 2, line 28: It wasn't quite clear to me what you meant by 'greenhouse ozone-depleting gases'. I guess you mean the CFCs and HCFCs? Gases with a non-zero ODP *and* non-zero GWP? It is an unfamiliar term (to me) and so maybe you want to consider using something better known.

We altered it to "ozone-depleting substances".  These happen to be CFCs, N2O, and CH4, which are GHGs, but the relevance here is that they are ODSs

Page 3, line 5: I think you should be more specific and say which 'other model'.

The sentence was revised to be "In the first use of Linoz in E3SMv1, the O3v1 module prescribed tropospheric ozone based on decadal monthly zonal mean latitude-by-pressure data from the input4MIPS Ozone data set v1.0 (Hegglin et al., 2016), and calculated stratospheric ozone interactively with Linoz v2."

Hegglin, M., Kinnison, D., Lamarque, J.-F. and Plummer, D.: CCMI ozone in support of CMIP6 - version 1.0, doi:10.22033/ESGF/input4MIPs.1115, 2016.

Page 3, line 8: It wasn't clear to me what you meant by 'these errors were not symmetrical'. Presumably the climatology overwrite would also place low tropospheric concentrations of ozone into the stratosphere? But what do you mean by 'these errors were not symmetrical'?

The errors were asymmetrical because of the implementation of the two chemical regimes and also because the ozone concentration is not linear across the tropopause:  it is almost constant in the upper troposphere, but increase almost exponential in the lower stratosphere.  When the model tropopause rises above the climatological tropopause, the tropospheric air between the two tropopauses is reset to have stratospheric ozone values 2 to 4 times larger than the tropospheric values and hence the errors.  When the model tropopause descends below the climatology, the stratospheric air between the two tropopauses will be handled by Linoz and thus no overwriting errors.

Page 4, line 10: Can you please cite a few papers that support the assertion that 'This model has proven robust and reasonably accurate'.

We added the citations to Déqué et al., 1994, McLinden et al., 2000,  and Eyring et al., 2013.

Déqué, M., Dreveton, C., Braun, A. and Cariolle, D.: The ARPEGE/IFS atmosphere model: a contribution to the French community climate modelling, Clim. Dyn., 10(4), 249–266, doi:10.1007/BF00208992, 1994.

McLinden, C., S. Olsen, B. Hannegan, O. Wild, M. Prather, and J. Sundet (2000) Stratospheric ozone in 3-D models: a simple chemistry and the cross-tropopause flux, J. Geophys. Res., 105, 14653-14665.

Eyring, V. et al : Long-term ozone changes and associated climate impacts in CMIP5 simulations, J. Geophys. Res. Atmospheres, 118(10), 5029–5060, doi:https://doi.org/10.1002/jgrd.50316, 2013.

Page 4, line 13: I guess then that what you really want is a tropopause-indexed ozone climatology e.g. Sofieva, V.F.; Tamminen, J.; Kyrölä, E.; Mielonen, T.; Veefkind, P.; Hassler, B. and Bodeker, G.E., A novel tropopause-related climatology of ozone profiles, Atmospheric Chemistry and Physics, doi:10.5194/acp-14-283-2014, 2014?

Yes, the Sofieva et al. tropopause indexing would solve the problems at the tropopause, but it would likely be a pain to implement safely in a climate model.  For example, if one grid cell is pushed up, how far "up" does the push reach?  If it were shifted independent of neighbors, then a differential heating would occur across neighbor cells at a higher altitudes and induce an artificial residual circulation.  Could be fun to try this, but Linoz is simpler we think, and also it can respond to changes in the ODSs.

Page 5, line 6: 'lower boundary sink' of ozone presumably?

We added ozone to the sentence.

Page 5, line 11: Again it is not clear to me what you mean by 'set to match the observed Antarctic ozone'? Do you mean that tunable parameters in the O3v1 module were set so that simulations of stratospheric ozone using this module would replicate the characteristics of the observed Antarctic ozone hole? If that is what you mean, perhaps that's what you should write.

Thank you, the text was revised as suggested.

Page 6, line 12: Is it driven by forecast winds or reanalysis winds?

We revised the sentence to clarify: "The UCI CTM is driven by 24-hour forecasts that were initialized with observationally assimilated data and spun-up for 12 hours, and thus …"

Page 6, line 13: I know what you mean by 'time-specific observations' but other readers may not. Perhaps better to say 'the true state of the atmosphere rather than a state

with the same climate but different weather as would be the case with E3SM'.

Thanks for pointing out this potential clarification issue, but we feel it is not difficult for readers to understand the meaning of "time-specific observations" within the context. We have improved the opening of this sentence (above).

Page 7, line 11: Have the MERRA TCO data been validated? e.g. do you see a clear discontinuity when you go from measured values to MERRA-filled values in any daily TCO field?

No, this extended dataset is used heavily in the ozone assessments and extensively analyzed. We don't see a clear discontinuity in our analysis.

Page 7, line 13: The minimum TCO over what geographic domain?

We added "in the southern hemisphere" to the line.

Page 7, line 21: What is 'milli-cm-Amagats'? I have never seen that before. Wouldn't "1DU = 2.69 x 10^16 molecules/cm2)" make more sense to more people?

The Dobson Unit was defined in terms of Amagats, a standard physical unit. The ozone community knows it well. The Dobson Unit is not a standard physical measure. We see no reason to introduce unnecessary numbers. From Wikipedia:
"An *amagat* is a practical *unit* of number density. Although it can be applied to any substance at any conditions, it is defined as the number of ideal gas molecules per *unit* volume at 1 atm (= 101.325 kPa) and 0 °C (= 273.15 K)."

Page 8, line 6: Couldn't you also patch the SCO fields with MERRA ozone data or are the MERRA ozone data not vertically resolved?

We could patch the MERRA SCO data, but for the global picture (not the Antarctic ozone hole diagnostics) it is much better to rely on the direct satellite observations of SCO. The range here covers 87% of the globe. While we need to use MERRA SCO for ozone hole comparisons and the community does so also, the community usually looks at global ozone like we do here because: 1) there are uncertainties from the assimilation model and they change with time; 2) the quality of assimilation data is constrained by the quality of observations assimilated into the model, which are also from MLS and OMI after year 2004 (Table 1 of Wargan et al., 2017), the same observations as in our study. Therefore, we don't see clear benefits of including MERRA SCO.

Wargan et al. (2017), Evaluation of the Ozone Fields in NASA's MERRA-2 Reanalysis, J. of Climate, 2961–2988.

Page 8, line 12: I would refer to these as 'biases' rather than 'errors'.

Changed.

Page 8, line 12: What, exactly, is 'excellent in the tropics but too great at high latitudes'?

Thanks. We changed it to "EAM versions have excellent seasonal phase and magnitude in the tropics but too great magnitudes at high latitudes." to be clearer.

Agreed. Especially over the southern hemisphere – text revised.

Page 8, line 20: By 'STD' I assume you mean standard deviation? I think you need to state that more clearly.

We defined standard deviation (STD) in earlier versions, but it was overlooked during the editing. We added it back. Thanks for the catch.

Page 8, line 26: In what way is this 'peak QBO-like variability'. You are not showing anything with a quasi-biennial oscillation time scale.

This peak appears to be related to the QBO as revealed by our additional analysis (not included here to stay focused on the main topics of this paper). We deleted "QBO-like".

Page 8, line 28: It is not clear to me at all that 'STD/SCO provides a second test of the overall stratospheric circulation'. I think that you need to demonstrate that far more robustly. It is certainly not self-evident.

In the earlier portion of this paragraph, we mentioned that the interannual variability quantified by the STD/SCO was associated with the QBO and wintertime polar variations. Such variabilities cannot not be examined by the SCO monthly climatology. It is quite clear that this interannual variability is transport driven.  We rephrased this to:
" The STD/SCO provides a test of the interannual variability in the stratospheric circulation. "

Page 9, line 7: Seeing the phrase 'is probably adequate' in a paper does not fill me with confidence. Can't you do the statistical test and make a definitive statement?

Yes, this was a bit overwritten and confusing.  We have revised the text here (and shortened the UCI CTM analysis following).
" The decadal variability of this QBO-like interannual variability is seen for different periods of the same model that differ by up to 20% in STD.  We cannot expect agreement with observations for a climate simulation to be better, and thus both O3v1 and O3v2 can be considered a match. "

Page 9, lines 7-8: It wasn't clear to me what you meant by 'The jump in the longterm

UCI CTM STD'? Do you mean the change in standard deviation between the two periods for which the UCI CTM standard deviations were calculated?

Yes, that's what we meant. We revised the sentence to read as "The large pre-OMI to post-OMI shift in UCI STDs …".

Page 9, line 8-9: I have never known a switch in satellite data to cause changes in the wind-driven circulation! That would be very impressive. I think that you need to be much clearer in saying what you mean.

We have revised this and it is clearly speculation without further analysis, but this problem has been shown by others for the MERRA-2 fields, from another paper:

"Douglass et al. (2017) showed that the MERRA-2 transport circulation had time-dependent biases prior to the Aura period that caused poor agreement with simulated long-lived trace gases. Stauffer et al. (2019) showed that changes in the observing system in 1998 led to significant improvements in the MERRA-2 stratospheric circulation."

We have added the references:

Douglass, A. R., Strahan, S. E., Oman, L. D., & Stolarski, R. S. (2017). Multi-decadal records of stratospheric composition and their relationship to stratospheric circulation change. *Atmospheric Chemistry and Physics*, *17*(19), 12081–12096. https://doi.org/10.5194/acp-17-627 12081-2017

Stauffer, R. M., Thompson, A. M., Oman, L. D., & Strahan, S. E. (2019). The Effects of a 1998 Observing System Change on MERRA-2-Based Ozone Profile Simulations. *Journal of Geophysical Research: Atmospheres*, *124*(13), 7429–7441. https://doi.org/10.1029/2019JD030257

Page 9, lines 18-19: I don't understand the sentence 'The UCI CTM scores slightly better because of the high-latitude SCO' at all. First better than what? And second 'because of the high-latitude SCO' is not an explanation for anything.

For clarification, we rephrased the sentence to "The UCI CTM scores slightly better than EAM versions because of its superior representation of the high-latitude SCO".

Page 9, line 25: Does not always perform better than what?

We have revised and cleaned up this sentence:
"For example, case [2] may highlight errors in using a pieced-forecast meteorology (UCI) as opposed to one from a continuously solved dynamical core (EAM).

Figure 4: I think it would be worth stating in the figure caption over what period these climatologies were calculated.

We included the periods in the figure caption as suggested. Thanks.

Page 10, lines 9-10: I don't think the sentence 'This metric has been a standard test for 2D and 3D stratospheric chemistry models for decades' is necessary. Let's say it had only ever been used once before. Would that make your analysis any less appropriate?

Good point.  This sentence was deleted.

Page 10, line 10: While I could take a guess, it wasn't entirely clear to me what you meant by 'The model goal'. I think you should describe what you mean more clearly so that the reader doesn't have to guess.

Yes, fixed it:
" The goal in terms of matching the ozone profiles (not usually quantified) was to get peak ozone above 10 ppm at 10 hPa in the tropics and the slightly upturned contours (i.e., at 5 hPa the 6 ppm contours extend over a wider latitude range than at 20 hPa). "

Page 10, line 13: In what way are the seasonal upward shifts in the contours in the winter 'odd'? To me, they look entirely as you would expect.

The word "odd" was a mistake and was removed.

Page 11, line 11: I have no idea what you mean by 'This metric is a tough one'? Tough like Sylvester Stallone or tough like Arnold Schwarzenegger? I was also confused by 'but we will need to add some other models to see how well it works outside of Linoz chemistry'. OK then go and add more models if that's what you need.

Changed to "This metric is a difficult one for the models to have high scores" to keep the tough men out of the picture. We will add the results with non-Linoz chemistry, when they become available in the future EAM versions (under development now).

Page 11, line 17: This is somewhat true. Equivalent Effective Antarctic Stratospheric Chlorine increased quite a bit from 1990 to 2000 and then decreased more slowly thereafter.

OK, have added:
"but always been above the threshold for creating an ozone hole within the winter vortex.".

Figure 5: I am surprised that you are using minimum TCO as a metric when Müller, R.; Grooß, J.-U.; Lemmen, C.; Heinze, D.; Dameris, M. and Bodeker, G.E., Simple measures of ozone depletion in the polar stratosphere, Atmospheric Chemistry and Physics, 251-264, 8, 2008, warned against using it.

Yes, this is potentially a problem for several reasons and a lowest 10th percentile would be more robust. There can be deep mini-holes that are meteorologically driven (as in NH) and the models can have spurious numerical tracer noise. Nevertheless, it is one of the metrics adopted by the NASA Ozone Watch, where the long-term ozone hole observational data are archived and updated daily to present. We intended to be consistent with the NASA Ozone Watch data and believe it is reasonable to apply this metric given the scope of this study. Note that the other metric – area below 220 DU – is also used alongside it.

Page 11, line 22: Sorry by 'ozone column' do you now mean SCO or TCO?

We meant SCO (revised the text accordingly).

Page 12, lines 6-7: So why wasn't O3v1 tuned with a better PSC temperature threshold?

If we had a more flexible EAMv1 releasing deadline, we would have the opportunity to fine tune the PSC T threshold for O3v1. Fortunately, we now can do so for the O3v2 module. Also, O3v1 was tuned at some level (the PSC temperature changed from the original one in the UCI distribution) but when it produced a regular ozone hole, it was deemed sufficient.

Page 12, line 10: I suspect you mean Figure 3d here?

Good catch. Thanks.

Page 12, lines 10-13: It is not clear to me what you mean by the 'dynamical conditions'...' remain relatively isolated from the ozone hole chemistry'? Dynamical conditions play a huge role in the efficacy of ozone depletion chemistry in the Antarctic stratosphere. That's what accounts for all of the interannual variability in Antarctic ozone depletion.

We roughly separated the dynamical conditions into 2 categories here: related or unrelated to the O3v1-O3v2 changes near the tropopause. By this classification, the large-scale dynamical conditions are unrelated to the O3v1-O3v2 changes, and thus remain very similar between the O3v1 and O3v2 simulations and doesn't cause differences between the O3v1 and O3v2 ozone holes.

Page 12, lines 13-15: I don't understand the purpose, meaning, or relevance of these last two sentences. Unless you have compelling reasons not to, I would suggest just deleting them.

One sentence is an important simple description creation of the ozone hole and is moved earlier. The other is dropped.

Page 12, line 22: By 'STE flux' do you mean the flux in general (i.e. kg/m^2/sec) or do you mean the ozone flux specifically?

Modified to "STE ozone flux".  As noted by others, the STE flux of air is not a well-defined quantity.

Page 13, lines 3-4: There is no place for a sentence like this is a paper. Either you did collect 'enough different models with enough similar results' to build a Taylor diagram or you didn't. So which is it? Otherwise what are you hoping for the reader to conclude from this sentence? It seems like speculation with no purpose.

Yes, good point. This sentence was deleted.

Page 13, line 6: I find this sentence very confusing. What, exactly, is set to the lowest four layers? The tropospheric ozone loss? But why would that be quantitatively equivalent to the STE ozone flux? Maybe I am misunderstanding something here? But if I am, it is possible that other readers would too. I think that you need to explain yourself much more clearly here.

We have rewritten this paragraph to be clearer and simpler:
"In O3v2 the net STE ozone flux is calculated from the loss in the near-surface (lowest 4) atmospheric layers.  Ozone is conserved in the rest of the troposphere and so the STE flux is taken up by these lowest layers.  It is resolved geographically and monthly, but because of the tropospheric transport from tropopause to lowest layers, the STE ozone flux diagnosed this way will differ from the tropopause-crossing flux in location and with a slight time delay, less than a month (Jacob, 1999).  In the UCI CTM, the STE flux is diagnosed at the tropopause as defined by an e90 tracer (Prather et al., 2011) and is able to resolve the STE fluxes across multiple tropopauses in the same column (Hsu et al., 2005; Tang et al., 2013; Hsu and Prather, 2014). Near-surface uptake of O3v2 ozone is minimal in the tropics, and thus we compare these two modelled STE fluxes as monthly hemispheric means."

Page 13, line 7: Do you really mean 'averaged over latitude and month'? So you have a monthly mean for every longitude? I wouldn't understand why you would do that.

Actually it is monthly by geographic grid cell.  However, given the time from tropopause to surface, a zonal mean, or hemispheric mean (here) is the best comparison.

In some of our previous studies using the UCI CTM, we averaged the STE ozone flux over latitude and month (e.g., Fig. 10 of Hsu et al., 2005) or over latitude, longitude, and month (e.g., Fig. 8 of Hsu et al., 2005) when it's important to resolve the flux geographically. Nevertheless, the current EAM version with O3v2 doesn't have such a capability to calculate the STE ozone flux maps.

Hsu, J., M. J. Prather, and O. Wild (2005), Diagnosing the stratosphere-to-troposphere flux of ozone in a chemistry transport model, J. Geophys. Res., 110, D19305, doi:10.1029/2005JD006045.

Page 13, line 10: But how are you getting zonal means when you averaged over latitude (I am assuming you meant averaged over all latitudes)?

Revised to "monthly hemispheric means".

Page 13, line 17-18: Replace 'peaks in May and bottoms in Dec' with 'maximizes in May and minimizes in December'.

Done.

Page 13, line 18: Regarding 'the peak extends to Jun'. Here and throughout, there is no need to use abbreviations for months in the manuscript text. That extra 'e' isn't going to blow out your publications budget.

We changed all the months to full names.

Page 14, line 4: Are you going to be examining the changes between O3v1 and O3v2 in greater detail (i.e. digging into how the coding of O3v1 and O3v2 differs) or are you going to be examining how changes from O3v1 to O3v2 affect the distribution of ozone in the UT/LS etc.? My primary complaint about this paper is that your are being too vague in your writing and it is often not clear exactly what you mean.

Apologies, we fixed to "the ozone changes" in the sentence to make it clear that this session is about the results changes, not the code changes.

Page 17, line 21: I don't understand what you mean by 'defined proportional to the reverse of lambda'. Do you mean the inverse of lambda?

That's right. Revised "(defined as being proportional to the inverse of $\lambda$)". Thanks.

Page 18, line 10: Delete the sentence 'Running these metrics with O3v1, O3v2, and the UCI CTM was informative.' If this wasn't the case you shouldn't have written the paper so to some extent it is self-evident.

This sentence was removed as suggested. Good point.

Page 18, line 14: In what way is the temperature threshold for PSC formation 'delicate'?

"extremely delicate" was deleted from that sentence.

Page 19, line 10: I would suggest informal editorial comments such as 'So we must accept good fortune:' should have no place in a paper. Unless of course you can cite a paper, perhaps from the humanities, that supports the assertion that 'we must accept

good fortune'.

Alright. This sentence was deleted.  Serendipity gets so little credit these days.

Page 19, line 15: I don't like to see phrases such as 'they seem to have much less impact on the fidelity' in a paper. Either it does impact the fidelity or it doesn't. Do a test so that you can state categorically which it is.

Revised to "they have much less impact on the fidelity" as that's what we see from the analysis presented here.

GRAMMAR AND TYPOGRAPHICAL ERRORS
I understand that the author's first language may not be English. It is not for me to say, but perhaps the second author could wordsmith the paper? To meet the standard of writing required for this journal, the quality of the writing needs to be improved, unless the journal employs a copy editor to do so. The list of corrections I have listed below is not complete and in some cases may only reflect my own writing style. They should be taken as suggestions.

We thank the reviewer for the detailed suggestions, which improve the quality of the paper, and apologize for the slack proofing.

Page 1, line 20: I think it would be clearer to say 'reduced bias' rather than 'shows improved mean bias'.

Done.

Page 2, line 15: Replace 'adopt mean climatological distribution' with 'adopt a mean climatological distribution of ozone'.

Changed. Thanks.

Page 3, line 1: Replace 'CTM' with 'CTMs'.

Done.

Page 3, line 16: Should this be 'E3SMv1' rather than just 'E3SM'? Likewise on line 23. Please ensure consistency in nomenclature throughout.

We made the changes consistently in the paper.

Page 4, line 23 and elsewhere: I would suggest you use the word 'shortcomings' rather than 'problems' in this context.

We feel these O3v1 weaknesses are problematic, so: "this problematic approach"

Page 4, line 24: Replace 'it is removed' with 'where it is removed' otherwise the grammar is wrong.

Thanks. Corrected.

Page 5, line 19: Do you mean 'forced only with observed SSTs'?

No, it is not forced only with observed SSTs. We now see the potential confusion and modified to read as "As a global climate model using observed SSTs as a lower boundary condition,"

Page 5, line 22: Delete 'check'.

Deleted.

Page 5, line 26: I would suggest shifting this sentence starting 'It would be interesting...' to the conclusions section where you might provide an 'outlook' of how this research could be extended.

This sentence was moved to the end of the first paragraph of the conclusion session.

Page 6, line 6: You have already defined the SST acronym earlier. You don't need to define it again.

Done.

Page 6, line 10: Replace 'through the end' with 'through to the end'.

Done.

Page 6, line 15: Replace 'are carried' with 'is carried'.

Changed.

Page 7, line 1: I would suggest 'tropospheric ozone column' rather than 'tropospheric column ozone'.

Both "ozone column" and "column ozone" are used in literature. We used "column ozone" consistently in this paper and would like to keep it that way.  Putting nouns together is fraught and we only seek consistency.

Page 7, line 8: Replace 'on the NASA's' with 'on NASA's'.

Revised.

Page 8, lines 21-22: This sentence is grammatically incorrect.

Revised to "presumably related to wintertime polar variations".

Page 9, line 15: Replace 'suggesting well-captured annual' with 'suggesting a well simulated annual'.

Added "a" after "suggesting", as suggested.

Page 10, line 3: Replace 'as diagnostic of chemistry' with 'as a diagnostic of chemistry'.

Done.

Page 11, line 26: 'frequent minimal values like 1995 and 2001' is worded poorly and should be rephrased.

We rephrased it to "more frequent occurrence of extreme ozone minima as in years 1995 and 2001."

Page 14, line 12: Replace 'at the lower stratosphere' with 'in the lower stratosphere'.

Changed.

Done.

Yes.

Changed.